# Hippocampal firing fields anchored to a moving object predict homing direction during path-integration-based behavior

Maryam Najafian Jazi[1], Adrian Tymorek[1], Ting-Yun Yen[1], Felix Jose Kavarayil[1], Moritz Stingl[1], Sherman Richard Chau[1], Benay Baskurt[1], Celia García Vilela [1] & Kevin Allen [1] ✉

Homing based on path integration (H-PI) is a form of navigation in which an animal uses self-motion cues to keep track of its position and return to a starting point. Despite evidence for a role of the hippocampus in homing behavior, the hippocampal spatial representations associated with H-PI are largely unknown. Here we developed a homing task (AutoPI task) that required a mouse to find a randomly placed lever on an arena before returning to its home base. Recordings from the CA1 area in male mice showed that hippocampal neurons remap between random foraging and AutoPI task, between trials in light and dark conditions, and between search and homing behavior. During the AutoPI task, approximately 25% of the firing fields were anchored to the lever position. The activity of 24% of the cells with a lever-anchored field predicted the homing direction of the animal on each trial. Our results demonstrate that the activity of hippocampal neurons with object-anchored firing fields predicts homing behavior.

Navigation in most species depends on path integration and its interactions with external landmarks. Path integration refers to a process in which a neuronal network integrates self-motion information (i.e., vestibular information, optic flow, feedback from motor commands) to keep track of the current location of the animal[1–3]. Path integration not only makes it possible for an animal to navigate in environments in which landmarks are novel, unreliable, or absent, but it also contributes to the formation of cognitive maps[4–6]. Understanding the neural circuits and computations supporting path integration has become a clinically relevant objective, as path integration is one of the first cognitive functions to be affected during the early stages of Alzheimer's disease[7,8].

The ability of animals to navigate using path integration is typically assessed with homing paradigms[3,9]. The most common paradigm assessing homing based on path integration (H-PI) is the food-carrying task on a circular arena[10–17]. In this paradigm, an animal explores a circular arena to find a large food reward. After collecting the food, the animal returns to its starting point to consume the reward. This homing behavior has been observed in complete darkness and

blindfolded animals, indicating that animals integrate self-motion cues during the outward journey and use this information to guide their homing behavior. Lesions of the hippocampus, medial entorhinal cortex, parietal cortex, and retrosplenial cortex impair H-PI[12–14,18], highlighting the essential role of these brain regions in H-PI.

In the single study characterizing the activity of spatially selective neurons during H-PI, Valerio and Taube[19] showed that the drift in the preferred direction of thalamic head-direction cells predicts homing error of rats in the food-carrying task. The activity patterns of other spatially selective neurons during homing are still unknown. Hippocampal place cells are part of a neuronal circuit processing self-motion cues. For instance, place cells can remain spatially selective when external landmarks are removed[20–22], implying that a path integration process can control place cell activity[4,23–28]. One unresolved question is whether the activity of hippocampal neurons during an H-PI task reflects a path integration process operating in a unified world-centered reference frame that includes the start and end point of the animal journey[2,29]. Standard place cells recorded during random foraging are examples of hippocampal neurons firing in a world-

[1]Medical Faculty of Heidelberg Univericsity and German Cancer Research Center, Heidelberg, Germany. ✉e-mail: allen@uni-heidelberg.de

centered coordinate system[29,30]. They generate a stable spatial representation covering the entire environment. Alternatively, the activity of hippocampal cells during H-PI could be regulated by both a world-centered reference frame and additional task-relevant reference frames[24,31,32]. Multiple reference frames were observed in rats navigating to a variably located goal within an open field. Besides traditional place fields that fired at a constant location, a subset of hippocampal neurons fired at a set distance and direction from the variably located goal[24]. Thus, under some conditions, the activity of hippocampal neurons can be controlled by at least two reference frames. Whether hippocampal neurons during homing tasks are controlled by a single world-centered reference frame or multiple reference frames has yet to be determined.

Current H-PI tasks for freely moving rodents are not ideal for studying the activity of spatially selective neurons. For instance, on the food-carrying task, an animal typically performs fewer than ten trials per day[10,13,15,33], providing very few data points to link cell activity to behavior. Ideally, an animal would perform numerous trials and thoroughly explore a large environment so that the firing rate of the neurons in a large area can be measured. A second limitation of current path integration paradigms is that they require highly trained experimenters to intervene in every trial performed by the animal. An automated and scalable paradigm for H-PI would improve reproducibility and facilitate rapid screening of different animal models[34–36].

Here, we adapted a classic operant conditioning protocol[37] to address the limitations of current H-PI paradigms. On the AutoPI (*Auto*mated *P*ath *I*ntegration) task, a mouse leaves a home base to search for a lever located at a random position on an adjacent circular arena. After pressing the lever, the mouse returns to the home base to collect a small food reward. When the task is performed in darkness, the ability to return directly to the home base depends on path integration. Mice could complete more than 100 trials in a single session without human intervention. We characterized the spatial representations of hippocampal CA1 pyramidal cells in mice performing the AutoPI task. Several neurons had firing fields anchored to the position of the lever. In darkness, the spatial selectivity of these lever-anchored fields was reduced during trials associated with a large homing error. Notably, the activity of a subset of lever-anchored fields predicted the homing direction of the animal. These results highlight how firing fields anchored to a variably positioned and task-relevant object encode directional information that predicts homing direction.

## Results

The AutoPI task was performed on an apparatus that consisted of a home base, a circular arena, and an autonomous lever (Fig. 1a, see Materials and Methods section for details). The home base contained a food magazine where food pellets could be delivered. A cohort of 13 mice was first trained to press the lever in the home base to obtain food rewards (Supplementary Fig. 1, Supplementary Movie 1). The lever was then moved out of the home base and onto the arena over several days. On each trial, the mouse had to leave the home base, navigate to the lever on the arena, press the lever, and return to the home base to receive a food reward (Fig. 1b). The position and orientation of the lever changed between trials (Supplementary Fig. 2). Training was completed once the mice could reliably press the lever independently of its orientation or position on the arena (approximately 16–21 training days) and trials in darkness (dark trials) were introduced thereafter. Testing sessions started with seven light trials before alternating between light and dark trials (Fig. 1c). The session ended when 100 trials were completed or 60 min had elapsed.

On average, mice performed 71.45 trials during a test session (n = 67 sessions from 13 mice). We analyzed 4453 trials from 13 mice (2429 light and 2024 dark trials, Supplementary Movie 2, Fig. 1d, e). On a typical trial, the mouse would search for the lever box, run around the lever box to find and press the lever and return to the arena periphery in a relatively straight line (Fig. 1f). The mouse sometimes required several journeys on the arena to complete a trial. (Fig. 1g). A journey started each time the mouse moved from the bridge to the arena and ended when the mouse left the arena. A trial often contained several journeys if (1) the mouse did not find the lever box and returned to the bridge, or (2) it found the lever box but failed to press the lever before returning to the bridge. Mice performed more journeys per trial during dark trials (Fig. 1h), which resulted in trials with longer duration (Fig. 1i). In the dark, mice were less likely to find the lever on a given journey (Fig. 1j). However, once they found the lever, the probability of pressing the lever was similar during light and dark trials (Fig. 1k). Similar plots presenting the distribution of trials instead of the aggregate score per mouse are shown in Supplementary Fig. 3.

### Characteristics of the search path predicting homing error

Mice had different navigational behavior during light and dark trials (Fig. 2a). During light trials, the mouse typically followed straight trajectories from the bridge to the lever box (Fig. 2a). During dark trials, the mouse had to explore more of the arena to reach the lever box, which resulted in complex search paths (Fig. 2a). To quantify these differences in navigational behavior, we compared the search and homing paths of light and dark trials on journeys where the mouse pressed the lever. The search path started when the mouse entered the arena and ended when the mouse found the lever box. The homing path started when the animal left the lever box and ended when it reached the periphery of the arena. The portion of the path when the animal was at the lever box was excluded. As expected, the search and homing paths were longer and lasted longer during dark trials (Fig. 2b, c). The running speed for both search and homing paths was much lower during dark trials than during light trials (Fig. 2b, c). Homing paths during light trials were especially fast, with a mean running speed of 64.18 cm/s, more than double that observed during homing paths in darkness (31.58 cm/s)(Supplementary Fig. 4a, b). Search and homing paths of dark trials were generally more complex (see Methods) than those of light trials (Fig. 2b, c). In dark trials, homing paths were less complex than the search paths (Search-Dark = 2.23, dark-homing = 1.83; n = 13 mice, Wilcoxon signed-rank test, P = 0.013), indicating that the mice returned to the arena periphery using straighter trajectories. The distributions of trials instead of aggregate score per mouse are presented in Supplementary Fig. 4a, b. Using the four features (length, duration, mean running speed, and complexity) of the search and homing paths, it was possible to predict whether a trial was performed in the light or in the dark with an accuracy of 0.91 (standard deviation: 0.012) (linear support vector machine model, 10-fold cross-validation).

To assess the ability of the mouse to return to the home base after pressing the lever, we calculated the error at the periphery in each trial (Fig. 2d, Supplementary Fig. 4c). Error at the periphery was significantly larger during dark trials compared to light trials (Fig. 2d, Supplementary Fig. 4d). The median error at the periphery was 4.26° and 18.15° for light and dark trials (Supplementary Fig. 4d), respectively. Importantly, homing error at the periphery in darkness was significantly lower than chance levels (chance levels = 90°, n = 13 Wilcoxon signed-rank test, $P = 2.4 \times 10^{-4}$). This indicates that mice were not heading randomly after pressing the lever during dark trials. Instead, they headed roughly toward the home base but with less accuracy than during light trials.

Path integration involves processing self-motion cues to estimate one's position and orientation. Because of the imperfect nature of sensory information and its integration, error in the position and orientation estimates is expected to increase with time and distance run from the start of a journey[11,38–40]. When visual landmarks are present, this error can be corrected using known landmarks[41]. During dark trials, error correction is less likely. Thus, in the AutoPI task, we expected that homing error correlates with

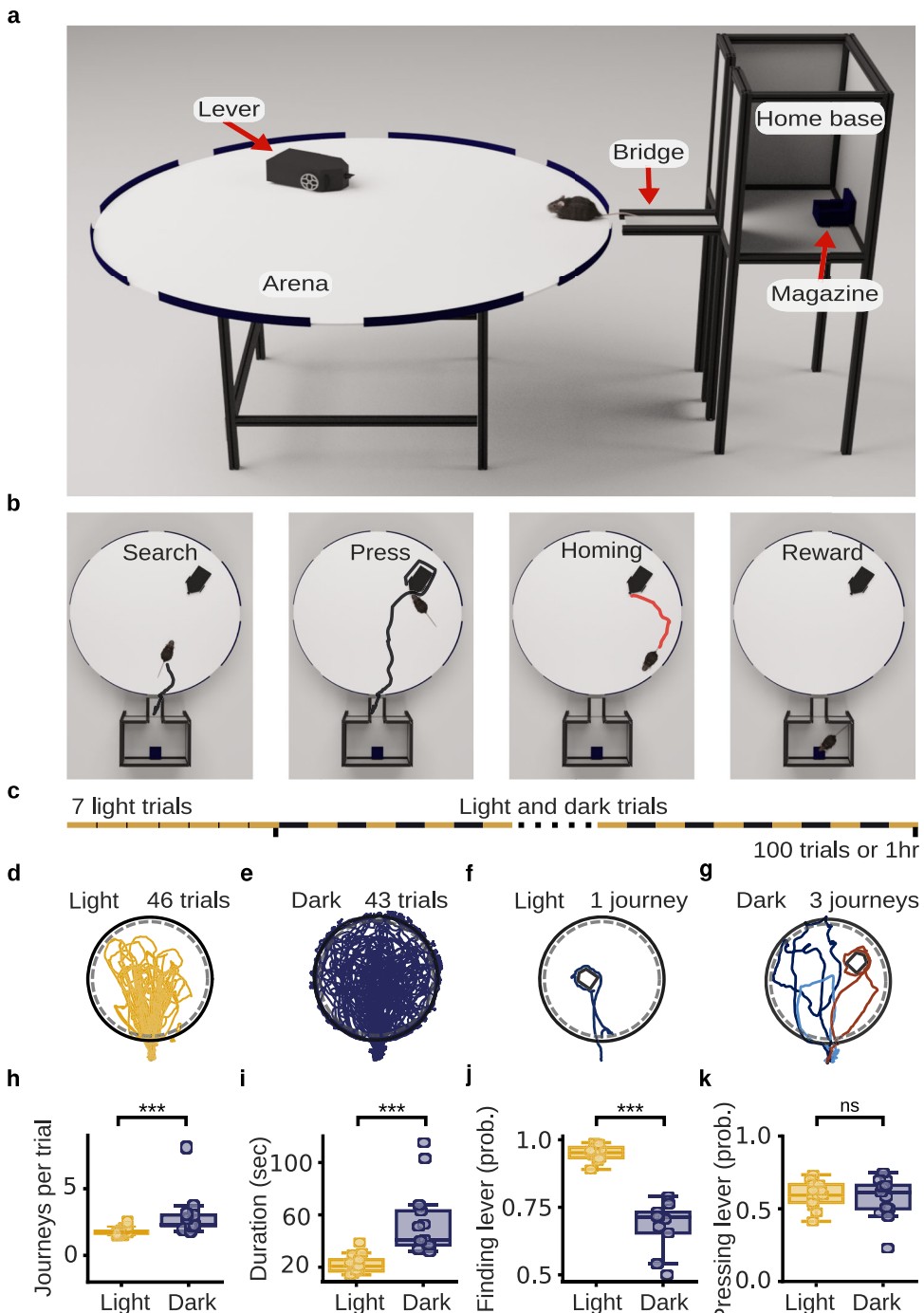

**Fig. 1 | Automated path integration task (AutoPI). a** The main apparatus consisted of a home base, a circular arena, and a lever. The home base contained a food magazine on the wall opposite to the arena. A motorized, inverted sliding door gave access to the bridge that led to the arena. The lever was built on a motorized platform, allowing the lever box to change its position and orientation between trials. **b** Four main phases of a trial on the AutoPI task. The animal leaves the home base to search for the lever on the arena (Search). The mouse then presses the lever (Press) and returns to the home base (Homing). A food reward is available in the magazine (Reward). **c** Schematic of a testing session. A session started with seven light trials, followed by a series of trials alternating between light and dark trials. The session ended after 100 trials or when 60 min had elapsed, whichever came first. **d** Running path for all light trials of a test session. The black circle and the dashed gray circle represent the edge of the arena and its periphery, respectively. **e** Running path for all dark trials from the same testing session shown in **d**. **f** Path of the mouse during a single journey of a light trial. **g** Path of the mouse during three journeys of a dark trial. The three journeys are plotted in different colors. **h** Number of journeys per trial for light and dark trials for each mouse (n = 13 mice, two-sided Wilcoxon signed-rank test, $P = 2.44 \times 10^{-4}$). **i** Trial duration for light and dark trials (n = 13 mice, two-sided Wilcoxon signed-rank test, $P = 2.44 \times 10^{-4}$). **j**, Probability of finding the lever on journeys associated with light and dark trials (n = 13 mice, two-sided Wilcoxon signed-rank test, $P = 2.44 \times 10^{-4}$). **k** Probability of pressing the lever once the lever had been found during light and dark trials (n = 13 mice, two-sided Wilcoxon signed-rank test, $P = 0.78$). ***$P < 0.001$, ns non-significant. Source data are provided as a Source Data file.

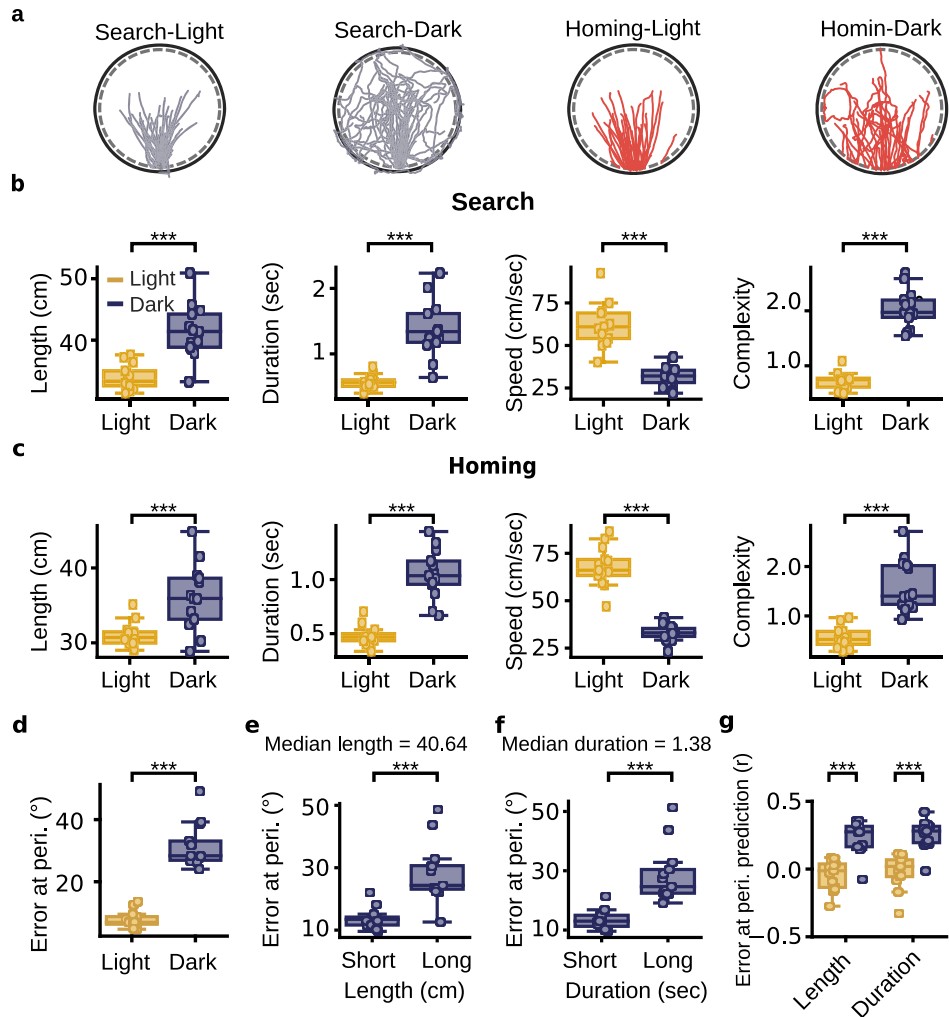

**Fig. 2 | Characteristics of search and homing paths during light and dark trials. a** Examples of search and homing paths during light and dark trials from one testing session. The search and homing paths for light and dark trials are shown separately. **b** The search path length (n = 13 mice, two-sided Wilcoxon signed-rank test, $P = 2.4 \times 10^{-4}$), duration ($P = 2.4 \times 10^{-4}$), speed ($P = 2.4 \times 10^{-4}$), and complexity ($P = 2.4 \times 10^{-4}$) for light and dark trials separately. Each small square represents one mouse. **c** The homing path length (n = 13 mice, two-sided Wilcoxon signed-rank test, $P = 2.4 \times 10^{-4}$), duration ($P = 2.4 \times 10^{-4}$), speed ($P = 2.4 \times 10^{-4}$), and complexity ($P = 2.4 \times 10^{-4}$) for light and dark trials separately. **d** Median error at the periphery for each mouse during light and dark trials (n = 13 mice, two-sided

Wilcoxon signed-rank test, $P = 2.4 \times 10^{-4}$). **e** Homing error at the periphery for dark trials with short and long search paths (n = 13 mice, two-sided Wilcoxon signed-rank test, $P = 4.8 \times 10^{-4}$). Dark trials were classified based on whether the search path length was below or above the median of all dark trials per mouse. **f** Same as in (**e**) but for search path duration (n = 13 mice, two-sided Wilcoxon signed-rank test, $P = 2.4 \times 10^{-4}$). **g** Correlation coefficients between homing error at the periphery and logarithmic transformation of path length (n = 13 mice, two-sided Wilcoxon signed-rank test, $P = 2.4 \times 10^{-4}$) and path duration ($P = 2.4 \times 10^{-4}$) of the search paths. Correlation coefficients are shown separately for light and dark trials. ***$P < 0.001$. Source data are provided as a Source Data file.

the length and duration of the search path of dark trials. We first divided the dark trials based on whether the length or duration of the search path was below or above the median score of all dark trials. Trials with larger search path length and search path duration were characterized by larger homing errors (Fig. 2 e, f). We also performed a correlation analysis between the characteristics of the search path and homing error at the periphery (Fig. 2g, Supplementary Fig. 4e). During dark trials, homing error at the periphery was positively correlated with search path length and duration (Fig. 2g). These significant correlations between the length as well as the duration of the search path and homing error support the hypothesis that mice used path integration to return to the home base during dark trials. Non-significant correlations for light trials could be explained by the presence of landmarks in the environment that prevented error accumulation in the path integration process.

We also addressed the possibility that mice used odors from objects outside the arena (e.g., home base, computer desk, etc.)

to return to the home base. To interfere with the use of odors, a fan located above the arena was used to create an airflow directed at the center of the arena. The airflow did not have a significant effect on the homing accuracy during dark trials (Supplementary Fig. 5, n = 6 mice, Wilcoxon signed-rank test, $P = 0.25$), suggesting that the homing behavior was not primarily controlled by odors originating from outside of the arena.

The results presented so far indicate that the AutoPI task allows the assessment of H-PI and that mice perform enough trials to characterize spatially selective hippocampal neurons. We next used the AutoPI task to test whether the neuronal activity of CA1 neurons is related to homing performance.

### Task-dependent hippocampal remapping
To characterize the activity of CA1 pyramidal cells during H-PI, we trained a second cohort of nine mice on the AutoPI task and implanted them with silicon probes targeted at the CA1 region. Recording sessions began with 30 min of random foraging on the arena, followed by

a 20-min rest period (Fig. 3a). The mouse then performed the AutoPI task for approximately 90–120 min. A second 20-min rest period concluded the recording session. Our analysis focused on 438 putative pyramidal cells recorded from the CA1 pyramidal cell layer of the hippocampus (Supplementary Fig. 6).

We first tested whether the spatial representations active on the circular arena during the AutoPI task were task-dependent[31]. We compared spatial representations active during the AutoPI task on the arena (all trials and paths combined) with the ones active during the random foraging trial. The AutoPI task and random foraging took place at the same location and on arenas of the same color, material, and dimensions. The only differences were that during random foraging, the 1.6 cm-high ridges at the arena's edge had no openings to escape the arena, and the home base was moved 30 cm away from the arena. We observed a significant reorganization of spatial representations when the mouse performed the AutoPI task (Fig. 3b–d and Supplementary Fig. 7). Neurons had very different firing rate maps when comparing the random foraging trial with the AutoPI task (Fig. 3b). To quantify hippocampal remapping, we considered all pairs of simultaneously recorded pyramidal cells (n = 5168 pairs). We calculated the similarity of firing rate maps for each pair during random foraging and during the AutoPI task, limiting the analysis to periods when the animal was on the circular arena (Fig. 3c). The similarity of firing rate maps was highly stable when comparing the first and second halves of the random foraging trial (Fig. 3c, n = 5168 cell pairs, RF1-RF2: $r = 0.66$, $P < 1.0 \times 10^{-20}$), or when comparing two independent sets of trials from the AutoPI task (n = 5168, A1-A2: $r = 0.53$, $P < 1.0 \times 10^{-20}$). In contrast, the map similarity was not significantly correlated when comparing random foraging and the AutoPI task (Fig. 3c, RF-AutoPI, n = 5168, r = 0.02, $P = 0.096$). These results indicate a near-complete reorganization of hippocampal spatial representations when mice performed the homing task.

We also estimate the stability of firing rate map similarity across conditions for pairs of neurons using mice as statistical units. Remapping between random foraging and the AutoPI task was observed in all mice with at least 10 cell pairs (Fig. 3d, RF1-RF2 Vs. RF-AutoPI). The remapping between random foraging and the AutoPI task was observed when we compared the random foraging to either the light or dark trials of the AutoPI task (Fig. 3d, RF1-RF2 Vs. RF-Light or RF1-RF2 Vs. RF-Dark).

Because the navigational demands differ between light and dark trials as well as between search and homing behavior of the AutoPI task, we tested whether different spatial representations were active during light and dark trials and during search and homing behavior[31,42,43] (Fig. 3e–g). We identified search and homing paths of both light and dark trials (Fig. 3e) and calculated firing rate maps during the different conditions. For this analysis, we focused on a region of the arena between the bridge and the arena center that was relatively well covered by the mice in the different conditions (Fig. 3e). The spatial firing pattern of most neurons showed very little resemblance between conditions (Fig. 3e), suggesting a reorganization of the spatial representations between light and dark trials and between search and homing behavior. To estimate remapping, we calculated firing rate map similarity for pairs of neurons and compared the stability of firing rate map similarity across conditions. Map similarity stability was generally higher when comparing the same behavioral condition across two independent sets of trials than when comparing across behavioral conditions (Fig. 3f). To statistically compared map similarity stability score, we calculated a stability score per mouse (Fig. 3g). We observed a significant decrease across light conditions and behaviors compared to the stability observed between two subsets of data from the same light condition or behavior (Fig. 3g). For instance, map similarity stability between search and homing behavior during light trials was lower than that for two sets of search epochs from light trials (Fig. 3g, SL-HL Vs. SL1-SL2, n = 8 mice, Wilcoxon

signed-rank test, $P = 7.81 \times 10^{-3}$). Similar patterns of results were obtained when assessing remapping via a population vector approach instead of analyzing cell pairs (Supplementary Fig. 7). These results indicate that the cell ensembles changed considerably from light to dark trials and search to homing behavior. Because of this remapping between light conditions and behaviors, we treated these different conditions separately in the following section.

## Distance to the lever box modulates hippocampal firing fields during search and homing behavior

We next characterized the firing fields of hippocampal neurons during search and homing behavior, excluding the time that the animal was near the lever box. In Fig. 4a, we plotted, for two neurons, the search path of every light trial together with the spikes and the y-position of the lever box on each trial. The first neuron fired at a fixed position independently of the lever position. In contrast, the second neuron appeared to fire at a set distance from the lever. To explore this phenomenon further, we generated traditional 2D firing rate maps and trial matrices for the search and homing behavior of light and dark trials (Fig. 4b). The trial matrices contained the firing rate of a neuron on each trial, one trial per row, making it possible to assess the reliability of the firing pattern across trials. In the trial matrices, the firing rate was plotted as a function of either the mouse's position along the y-axis coordinate (axis parallel to a vector passing through the center of the bridge and the center of the arena) or the distance between the mouse and the lever box. Figure 4b shows examples of neurons with firing fields during search and homing behavior of light and dark trials. We observed neurons with stable place fields on the arena (labeled Place fields). In the trial matrix, these neurons consistently fired at the same y-axis coordinate across trials. In addition, the firing fields of a subset of neurons were influenced by the distance to the lever box (Fig. 4b, labeled Lever distance fields). In the trial matrices, these neurons appeared to fire at a constant distance from the lever box, and they fired at a variable y-axis position.

We quantified the influence of the y-axis coordinates and the distance to the lever box on the firing patterns of neurons. We calculated the trial matrix correlation, which estimates how similar the firing patterns observed on different trials within one trial matrix are (Fig. 4c, Supplementary Fig. 8). Trial matrix correlations were calculated separately for light and dark trials and for search and homing behavior. Only neurons with a mean firing rate larger than 2.0 Hz in a given condition were considered. The proportion of active neurons (firing rate > 2.0 Hz) across different conditions, which ranged from 0.616 to 0.673, was not significant (number of active neurons Search-Light: 295; Search-Dark: 285; Homing-Light: 274; Homing-Dark: 270, Chi-square test, $P = 0.715$). During dark trials, trial matrix correlations from matrices based on the Y-axis coordinate were significantly lower during homing than during the search path (Fig. 4c, left). During homing behavior, trial matrix correlations based on the Y-axis were significantly lower during dark than light trials. When analyzing trial matrix correlation matrices based on the distance to the lever (Fig. 4c, right), we found lower correlations during search behavior in darkness, suggesting that coding for lever distance was weak during search behavior in darkness. Presumably, estimating the distance from the lever box during search behavior required direct visual contact with the lever box.

As a second measure of the influence of the y-axis coordinate and distance to the lever, we calculated firing rate histograms with the firing rate as a function of the y-axis coordinate or distance from the lever, separately for the four conditions (Search-Light, Search-Dark, Homing-Light, Homing-Dark). In these firing rate histograms, all trials of one condition were merged. Information scores were calculated from these histograms (Fig. 4d). When considering firing rate histograms with the y-axis coordinate of dark trials, we observed lower information scores during search behavior compared to homing

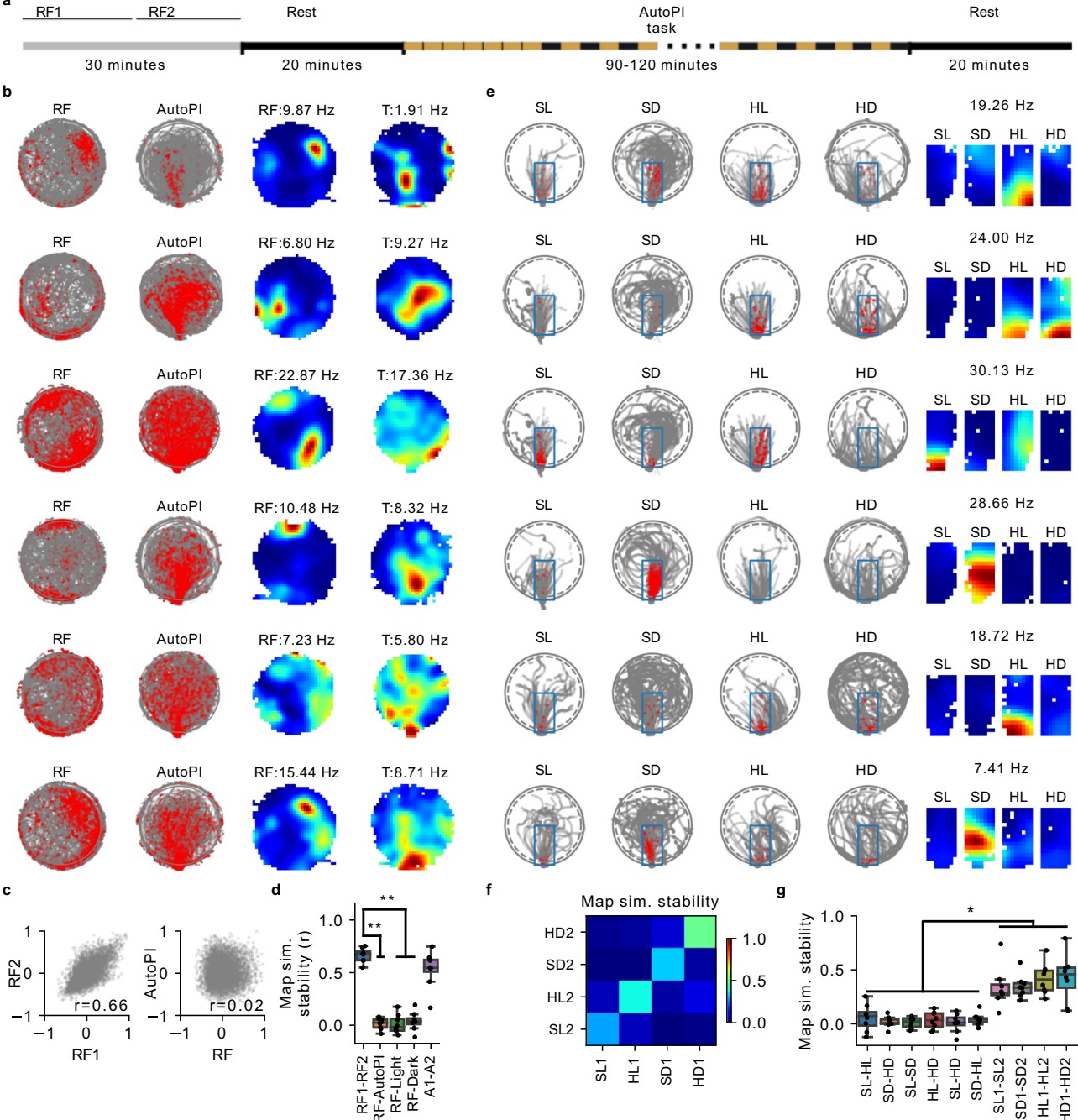

**Fig. 3 | Task-induced hippocampal remapping. a** Schematic of a recording session. The session started with a random foraging trial and a rest trial. The random foraging trial was divided into two halves (RF1 and RF2) in some analyses. The mouse then performed the AutoPI task for approximately 90–120 min, followed by an additional rest trial. **b** Examples of spike-on-path plots and firing rate maps during random foraging (RF) and the AutoPI task (T) for six neurons. The numbers above the maps indicate the peak firing rates of the neurons. **c** Left: firing rate map similarity for pairs of neurons (n = 5168) during the first and second half of the random foraging trial (RF1 and RF2). The correlation coefficient estimates the stability of the firing rate map similarity between the two conditions. Right: firing rate map similarity of pairs of neurons for random foraging trial (RF) and the AutoPI task. **d** Stability of maps similarity of pairs of neurons across different tasks and task conditions (first half of random foraging trial: RF1, second half of random foraging trial: RF2, all AutoPI trials: AutoPI, light trials of AutoPI task: Light, dark trials of the AutoPI task: Dark, two independent sets of AutoPI trials: A1 and A2) (RF1-RF2 Vs. RF-AutoPI: n = 8 mice, two-sided Wilcoxon signed-rank test, $P = 7.81 \times 10^{-3}$; RF1-RF2 Vs. RF-Light: $P = 7.81 \times 10^{-3}$; RF1-RF2 Vs. RF-Dark: $P = 7.81 \times 10^{-3}$). Each data point represents a mouse. **e** Spatial firing pattern of six neurons (different from those in **b**)

during the AutoPI task. Left: examples of spike-on-path plots during the AutoPI task. The data is shown separately for search (S) and homing (H) behaviors and for light (L) and dark (D) trials. The spikes are shown only for a zone of interest (blue rectangle) well covered in the four conditions (SL, SD, HL, HD). Right: firing rate maps of the zone of interest during the four conditions of the AutoPI task. The four maps are plotted using the same color scale, and the numbers above the maps indicate the peak firing rate of the neurons. **f** Matrix containing the stability of firing rate map similarity for pairs of simultaneously recorded neurons across four conditions (n = 5168 cell pairs). The trials of each condition were divided into two independent sets of trials (e.g., SL1 and SL2) to allow within-condition comparisons. **g** Stability of maps similarity of pairs of neurons across the four different AutoPI conditions (n = 8 mice, two-sided Wilcoxon signed-rank test, SL-HL vs SL1-SL2: $P = 0.0078$; SD-HD vs SL1-SL2: $P = 0.0078$; SL-SD vs SL1-SL2: $P = 0.0078$; HL-HD vs HL1-HL2: $P = 0.0078$; SL-HD vs SL1-SL2: $P = 0.0078$; SL-HD vs SD1-SD2: $P = 0.0078$; SL-HL vs HL1-HL2: $P = 0.015$; SD-HD vs HD1-HD2: $P = 0.0078$; SL-SD vs SD1-SD2: $P = 0.0078$; HL-HD vs HD1-HD2: $P = 0.0078$; SL-HD vs HD1-HD2: $P = 0.0078$; SD-HL vs HD1-HD2: $P = 0.0078$). *$P < 0.05$, **$P < 0.01$. Source data are provided as a Source Data file.

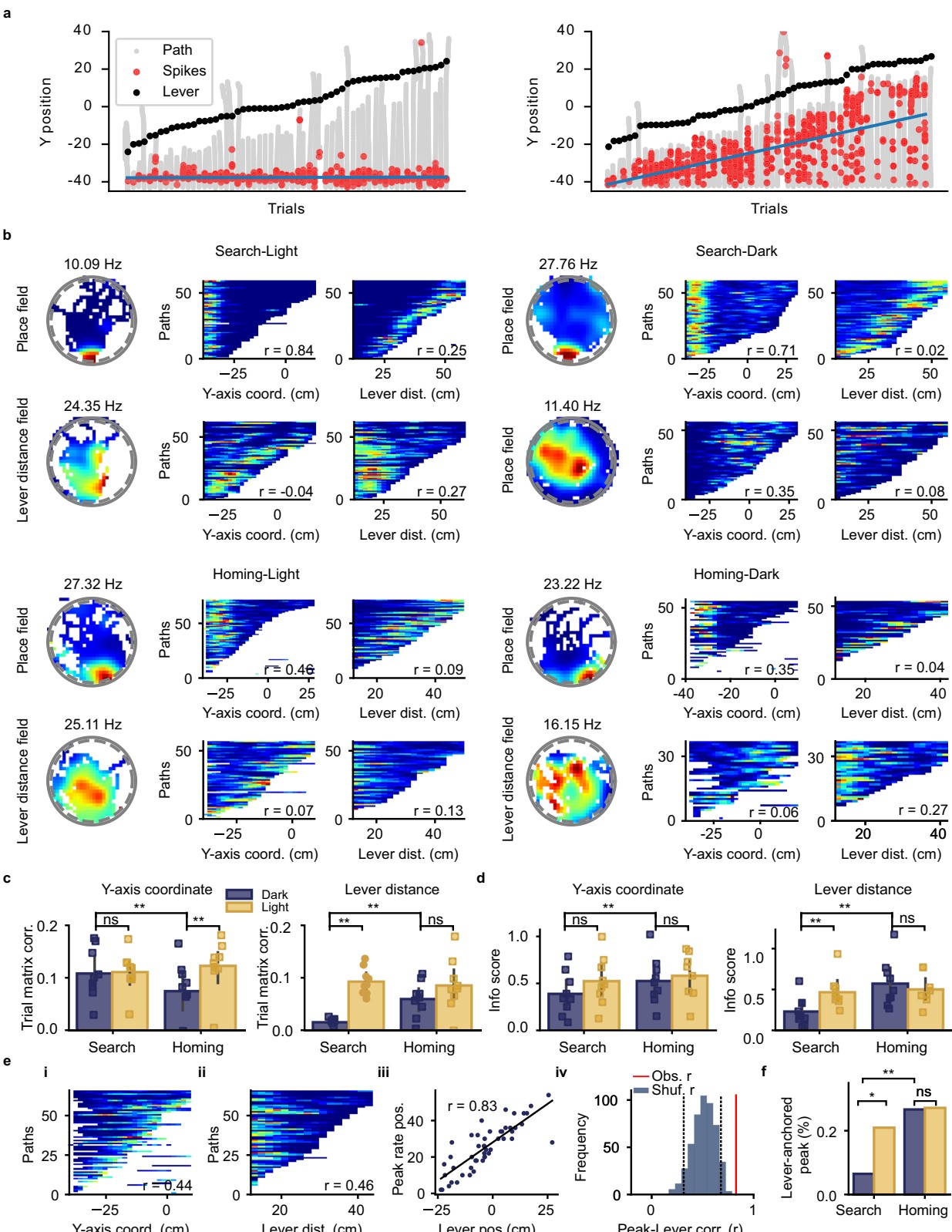

behavior (Fig. 4d, left). When considering information scores of histograms of the distance to the lever, we observed lower information scores while the animal searched for the lever in darkness compared to searching for the lever with visual landmarks (Fig. 4d, right). These results confirm that during dark trials, hippocampal neurons can encode information about distance from the lever box once the animal has encountered the lever box using path integration[24].

We next estimated the proportion of neurons that encoded information about the distance from the mouse to the lever box. We considered light and dark trials, and search and homing behavior separately and analyzed the trial matrices of individual neurons (Fig. 4e). In each condition, we tested whether the location of the firing rate peak in the y-axis coordinate (Fig. 4e i) was correlated with the y-axis coordinate of the lever box. Since this analysis required

**Fig. 4 | Firing fields of the search and homing paths modulated by the distance to the lever box. a** Example of two pyramidal cells firing during the search behavior of light trials. The neuron on the left fired as a function of position, while the neuron on the right fired as a function of the distance to the lever box. All search paths (gray lines) are shown next to each other and ranked according to the Y-position of the lever box (black dots) on each trial. The blue line is the regression line of the spikes (red dots) shown in the plot. **b** Example of pyramidal cells active during the Search-Light, Search-Dark, Homing-Light, and Homing-Dark conditions. Two different neurons are shown in each condition, one neuron per row. The first row in each condition shows a neuron firing at a set position (typical place cell, labeled as Place field). The second row in the Search-Light, Homing-Light, and Homing-Dark conditions are neurons that fired at a fixed distance from the lever (labeled as Lever distance field). For each neuron, the 2D firing rate map and two trial matrices are shown. The number above the 2D firing rate map is the peak firing rate in the map. The first trial matrix shows the firing rate as a function of the mouse's position on the y-axis (y-axis in the 2D firing map, 0 being the center of the arena and -40 the edge of the arena near the bridge). The second trial matrix shows the firing rate of the neuron as a function of the distance between the mouse and the lever box. The r value on each matrix is the trial matrix correlation. **c** Mean trial matrix correlation for active pyramidal cells obtained from trial matrices with y-axis coordinate (left) or distance to the lever box (right). The data are shown separately for light and dark trials and search and homing behavior (n = 9 mice, two-sided Wilcoxon signed-rank test with Bonferroni correction and $\alpha$ set to 0.0125, Y-coordinate, Search-Light vs Search-Dark: $P = 1$; Y-coordinate, Homing-Light vs Homing-Dark: $P = 3.9 \times 10^{-3}$; Y-coordinate, Search-Dark vs Homing-Dark: $P = 3.9 \times 10^{-3}$; Lever distance, Search-Light vs Search-Dark: $P = 3.9 \times 10^{-3}$; Lever distance, Homing-Light vs Homing-Dark: $P = 0.07$; Lever distance, Search-Dark vs Homing-Dark: $P = 3.9 \times 10^{-3}$); Data are presented as means +/- SEM. **d** Information score of 1D firing rate histograms with the firing rate as a function of y-axis coordinate (left) or distance to the lever box (right) (n = 9 mice, two-sided Wilcoxon signed-rank test with Bonferroni correction and $\alpha$ set to 0.0125, Y-coordinate, Search-Light vs Search-Dark: $P = 2.7 \times 10^{-2}$; Y-coordinate, Homing-Light vs Homing-Dark: $P = 0.12$; Y-coordinate, Search-Dark vs Homing-Dark: $P = 7.8 \times 10^{-3}$; Lever distance, Search-Light vs Search-Dark: $P = 3.9 \times 10^{-3}$; Lever distance, Homing-Light vs Homing-Dark: $P = 0.65$; Lever distance, Search-Dark vs Homing-Dark: $P = 3.9 \times 10^{-3}$); Data are presented as means +/- SEM. **e** Example of a neuron with a firing field influenced by the distance to the lever during the homing paths of dark trials. i and ii: Trial matrix of the neuron as a function of the y-axis coordinate and the distance to the lever box, respectively. iii: Y-axis location of the peak firing rate of the neuron and the y-axis location of the lever for single homing paths in darkness. The location of the peak firing rate of the neuron can be predicted by the lever box location. iv: Distribution of r values between firing rate peak location and lever location obtained after shuffling the firing rate values within each trial. The observed r value felt outside the random distribution, indicating a significant correlation between peak rate location and lever location. **f** Percentage of active pyramidal cells for which the location of their peak firing rate was significantly correlated with the lever position (Chi-square test, Search-Light Vs. Search-Dark: $P = 1.1 \times 10^{-2}$; Homing-Light Vs. Homing-Dark: $P = 0.94$; Search-Dark Vs. Homing-Dark: $P = 1.69 \times 10^{-3}$). ****$P < 0.0001$, ns: $P > 0.0125$. Source data are provided as a Source Data file.

identifying the peak firing rate of the neuron on single runs, we selected only neurons with an average peak firing rate above 7.5 Hz in the y-axis coordinate or in the lever box distance trial matrix and a significant y-coordinate or lever distance trial matrix correlation (number of neurons: Search-Light: 104; Search-Dark: 105; Homing-Light: 82; Homing-Dark: 73; chi-square test, $P = 0.03$). We plotted the peak firing rate location against the lever box location on the y-axis (Fig. 4e iii). If a neuron fired at a fixed distance from the lever box, there should be a positive correlation between the position of the peak firing rate and the position of the lever box (Fig. 4e iii). We calculated the r value between the location of the peak firing rate and the position of the lever box and compared it to a distribution of r values expected by chance (Fig. 4e iiii). Neurons with significant r values were classified as neuron encoding for the distance from the lever box.

During the search behavior, 22.11% and 8.57% of the selected neurons were influenced by the position of the lever box during light and dark trials (Fig. 4f). The percentage was significantly higher during light trials than during dark trials (number of neurons for Search-Light: 104, Search-Dark: 105; chi-square test, $P = 1.15 \times 10^{-2}$). The percentage of neurons encoding the distance to the lever box during the search path of dark trials was not significantly higher than that expected by chance (number of neurons for Search-Dark: 105, chance level of 5%, chi-square, $P = 0.4$). During homing behavior, 27.39% and 25.60% of the neurons were influenced by the position of the lever box during light and dark trials, respectively. The percentages during Homing-Light and Homing-Dark paths were not significantly different (number of neurons for Homing-Light: 82, Homing-Dark: 73; chi-square test; $P = 0.94$). During dark trials, the percentage of lever-distance encoding neurons was higher during homing than during search behavior (number of neurons in Search-Dark: 105, Homing-Dark: 73; chi-square test; $P = 1.6 \times 10^{-3}$). These results indicate that the firing field of approximately 25% of the reliably active neurons during homing behavior was influenced by the distance of the mouse from the lever box.

## Hippocampal firing fields anchored to the lever box

The previous analysis showed that some hippocampal neurons encoded the distance to the lever box during the search and homing path. We then focused on firing activity when the mouse was at the lever box. After reaching the lever box, a mouse typically ran a full circle around the lever box to find and press the lever. We hypothesized that some cells would consistently fire near the lever box, independently of the lever box position on the arena.

By overlying the spikes of single neurons on the path of the mouse during single trials, we discovered that several neurons fired action potentials when the mouse was around the lever (Fig. 5a). We calculated the firing rate of the neurons as a function of distance from the lever box. Several neurons showed a clear peak within 10 cm from the lever box (Fig. 5a). These peaks coincided with the distance around the lever box where the mice spent more time than for other distances. By plotting a trial matrix with a neuron's firing rate on every trial as a function of lever box distance, we found that many neurons were active at a given distance on most trials, independently of the lever box position (Fig. 5a). At a population level, we observed a significant accumulation of firing peaks near the lever during both light and dark trials (Fig. 5d, n = 438 peaks in firing rate histograms as a function of distance from the lever box, difference from a homogeneous distribution, $P < 0.01$).

The neurons with fields near the lever box fired only when the animal was in a specific direction from the lever box (Fig. 5a). To quantify the directionality of the firing fields, we transformed the position data so that the lever box center was always at position 0,0. This allowed us to generate lever-box-centered firing maps in which the lever box had a constant position. When calculating the direction of the mouse around the lever box, a reference vector needs to be set from which the direction is calculated. This vector, with its origin at the center of the lever box, sets the directional reference frame for the analysis. We compared the three directional reference frames illustrated in Fig. 5b. In the Cardinal reference frame, the reference vector was pointing south. In the Bridge reference frame, the reference vector was pointing toward the center of the bridge. In the Lever reference frame, the reference vector was pointing toward the lever (13 × 10 mm part of the lever box pressed by the mouse). Lever-box-centered firing rate maps and polar firing rate histograms were calculated in the three directional reference frames. These included only data from when the mouse was within 12 cm of the lever box. As shown in Fig. 5c, neurons were typically active only when the mouse was located in a specific direction around the lever box in the Cardinal and Bridge reference frames. Directionality appeared reduced in the Lever reference frame.

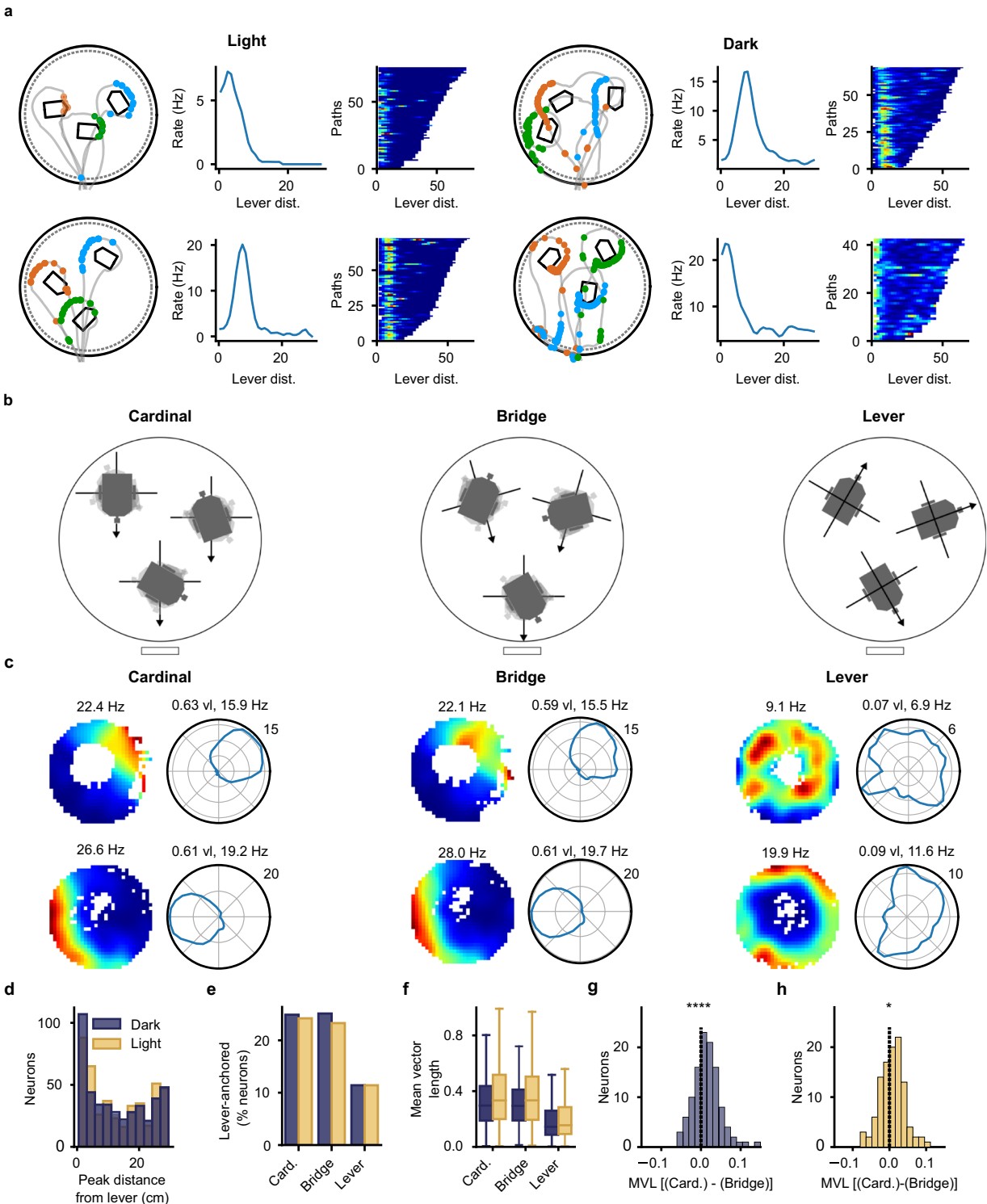

We identified cells with a stable firing field near the lever in the three directional reference frames. To be classified as a cell with a lever-box-anchored field, the neuron had to fulfill three criteria: (1) their peak firing rate as a function of lever distance was within 10 cm of the lever box, and this peak was significantly above the 95th percentile of a shuffled distribution, (2) their peak firing rate in the 2D lever-box-centered map had to be larger than 7.5 and (3) the similarity between two 2D lever-box-centered maps generated from two independent sets of trials within the same condition had to be larger than 0.4. Approximately 23–25% of the neurons had a lever-box-anchored firing

field within the Cardinal and Bridge reference frames (Fig. 5e). This percentage dropped to approximately 10–11% in the Lever reference frame. The percentage of lever-box-anchored neurons was statistically higher in the Cardinal and Bridge reference frames than in the Lever reference frame in both light and dark trials (n = 438 neurons, chi-square tests, all $P < 0.05$). There was no significant difference in the percentage of neurons with a lever-box-anchored field between the Cardinal and Bridge reference frames (n = 438 neurons, chi-square tests, all $P$ values > 0.05). There was also no difference in the percentage of lever-box-anchored neurons between light and dark trials in

**Fig. 5 | Hippocampal firing fields surrounding the variably placed lever box.**
**a** Examples of four neurons (two in light trials, two in dark trials, one per row) with firing fields active when the mouse runs around the lever box. Left: Spikes (dots) on the mouse paths (gray lines) during three trials. The spikes of the different trials are plotted using different colors. Middle: Firing rate as a function of distance from the lever box. Right: Trial matrix containing the firing rate of the neuron as a function of distance from the lever box. Each row of the matrix represents a single journey on the arena in which the animal pressed the lever. **b** Schematic depicting three possible directional reference frames when calculating the firing rate of a neuron around the lever box. Cardinal: Direction is calculated relative to a vector pointing South. Bridge: Direction is relative to a vector pointing towards the center of the bridge. Lever: Direction is relative to a vector starting at the center of the lever box and pointing towards the lever (part of the lever box pressed by the mouse). **c** Lever-box-centered firing rate maps and polar firing rate histograms in three different lever-box-centered reference frames (Cardinal, Bridge, and Lever). The maps and polar firing rate histogram include the data from all journeys in which the

mouse pressed the lever. The polar plot shows the firing rate of the neuron as a function of the direction of a vector originating at the lever box center and pointing towards the head of the mouse. The data in the first and second rows are from two different neurons and from light and dark trials, respectively. **d** Distribution of peak-firing-rate distance from the lever box. Many neurons had firing rate peaks close to the lever box. **e** Percentage of neurons with a firing field anchored to the lever box position. **f** Mean vector length of the lever-centered circular firing rate histogram in different reference frames (n = 348 neurons, two-way ANOVA on the mean vector length of the polar firing rate histogram; effect of *light*: $P = 3.17 \times 10^{-4}$; *Reference Frame*: $P = 2.19 \times 10^{-41}$). **g** Change in mean vector length of the lever-centered circular firing rate histogram between the Cardinal and Bridge reference frames for dark trials (n = 115 neurons, two-sided Wilcoxon signed-rank test, $P = 4.356 \times 10^{-6}$). **h** Same as (**g**) but for light trials (n = 113 neurons, Wilcoxon signed-rank test, $P = 3.0 \times 10^{-3}$). *$P < 0.05$, ****$P < 0.0001$. Source data are provided as a Source Data file.

any of the three reference frames (n = 438 neurons, chi-square tests, all $P > 0.05$). We calculated the preferred direction of each lever-box-anchored neuron from the polar firing rate histograms. The firing fields were homogeneously distributed around the lever box in all three reference frames (Number of cells in Cardinal-Light: 106, Cardinal-Dark: 109, Bridge-Light: 102, Bridge-Dark: 110, Lever-Light: 50 and Lever-Dark: 50; Rayleigh tests, all $P > 0.05$).

To compare spatial coding in the three reference frames, we compared the mean vector length of the polar rate histogram around the lever (Fig. 5f, Supplementary Table 1). A two-way ANOVA on the mean vector length of the polar firing rate histogram, including all 438 pyramidal cells, showed a significant effect of *Light* ($P = 3.17 \times 10^{-4}$) and *Reference Frame* ($P = 2.19 \times 10^{-41}$). The mean vector lengths in the Cardinal and Bridge reference frames were higher than those in the Lever reference frame ($P = 6.8 \times 10^{-15}$ and $P = 6.9 \times 10^{-13}$, respectively). These differences between reference frames were also observed when limiting the analysis to lever-box-anchored neurons.

The Cardinal and Bridge reference frames were often nearly aligned when the lever box was located near the x-axis center of the arena (Fig. 5b). Therefore, smaller differences are to be expected between these two reference frames. To contrast them more directly, we selected neurons that were classified as lever-box-anchored in the Cardinal and/or Bridge reference frames. We then performed a within-neuron comparison of their mean vector lengths in the two reference frames (Fig. 5g, h). We found larger mean vector lengths in the Cardinal than Bridge reference frames. This indicates that most firing fields located around the lever box encoded direction best in the Cardinal reference frame.

## Long search paths associated with reduced directional selectivity of lever-box-anchored firing fields
We found that several neurons had firing fields anchored to the lever box. We hypothesized that during dark trials, mice updated the estimate of their position and orientation using path integration and that error accumulation in the path integration process negatively affected the directional selectivity of lever-anchored firing fields[11,38–41]. To test this hypothesis, we separated dark trials into trials with short and long search paths (equally-sized groups) and assessed directional selectivity around the lever box for both groups. Figure 6a shows the firing activity of two neurons around the lever box for trials with short and long search paths. To quantify the directional selectivity around the lever box, we first calculated the mean vector length and peak rate of the lever-centered directional firing rate histograms for trials with short and long search paths. We observed higher mean vector length in trials with the short paths although the differences were not significant (Fig. 6b, n = 8, one-sided Wilcoxon signed-rank test, $P = 0.07$). The peak firing rates were higher during trials with short search paths (Fig. 6c). We used two additional measures to assess the stability of the

directional firing across trials. Firstly, we calculated trial matrices for trials with short and long search paths. These matrices contained the firing rate as a function of direction around the lever box for all journeys with a lever press. The stability of the directional firing rate across trials was assessed by calculating a trial matrix correlation (Supplementary Fig. 8), which assessed the similarity between the directional firing between trials. We also calculated the average direction trial drift from the trial matrices (Supplementary Fig. 9). We observed higher trial matrix correlations during trials with short search paths compared to trials with long search paths (Fig. 6d). Similarly, smaller mean trial drifts were observed during trials with shorter search paths (Fig. 6e). Similar results were observed when using search path duration instead of search path length (n = 8 mice, two-sided Wilcoxon signed-ranked test; MVL: $P = 7.8 \times 10^{-3}$, peak rate: $3.9 \times 10^{-3}$, trial matrix correlation: $7.8 \times 10^{-3}$, trial drift: $P = 0.15$). We also calculated correlation coefficients between the search path length and trial drift for the lever-box-anchored fields of each mouse (Fig. 6f). The correlation coefficients (mean per mouse) were significantly above 0 (Fig. 6g). These results demonstrate that longer search paths were associated with lower directional selectivity of lever-box-anchored fields, and provide evidence that the lever-box-anchored fields are driven by a path integration process.

## Instability of lever-box-anchored firing fields associated with inaccurate homing
Lever-box-anchored firing fields could reflect how the mouse perceived direction around the lever box, and this information could correlate with homing behavior. If so, we would expect a reduction in the directional stability of the lever-anchored firing fields in trials with inaccurate homing. We tested this by dividing the dark trials into two equally-sized groups that we labeled trials with accurate or inaccurate homing (Fig. 7a). Accurate and inaccurate homing were defined as homing with an error smaller or larger than the median homing error within each session, respectively. We estimated directional selectivity around the lever box from the mean vector length of the directional firing rate histograms and the trial matrix correlation (Fig. 7a, b). We found that the mean vector lengths and trial matrix correlations were larger for dark trials with accurate homing compared to dark trials with inaccurate homing (Fig. 7b). The mean firing rate of lever-box-anchored neurons was not significantly different during accurate and inaccurate trials (n = 8 mice, Wilcoxon signed-rank test, $P = 0.054$). Together, these findings indicate that the lever-box-anchored firing fields were more stable when the homing behavior of the mouse was accurate.

## The trial-to-trial direction of lever-box-anchored fields predicts homing direction during dark trials
We next tested the possibility that there might be a relationship between the firing direction of lever-anchored firing fields on

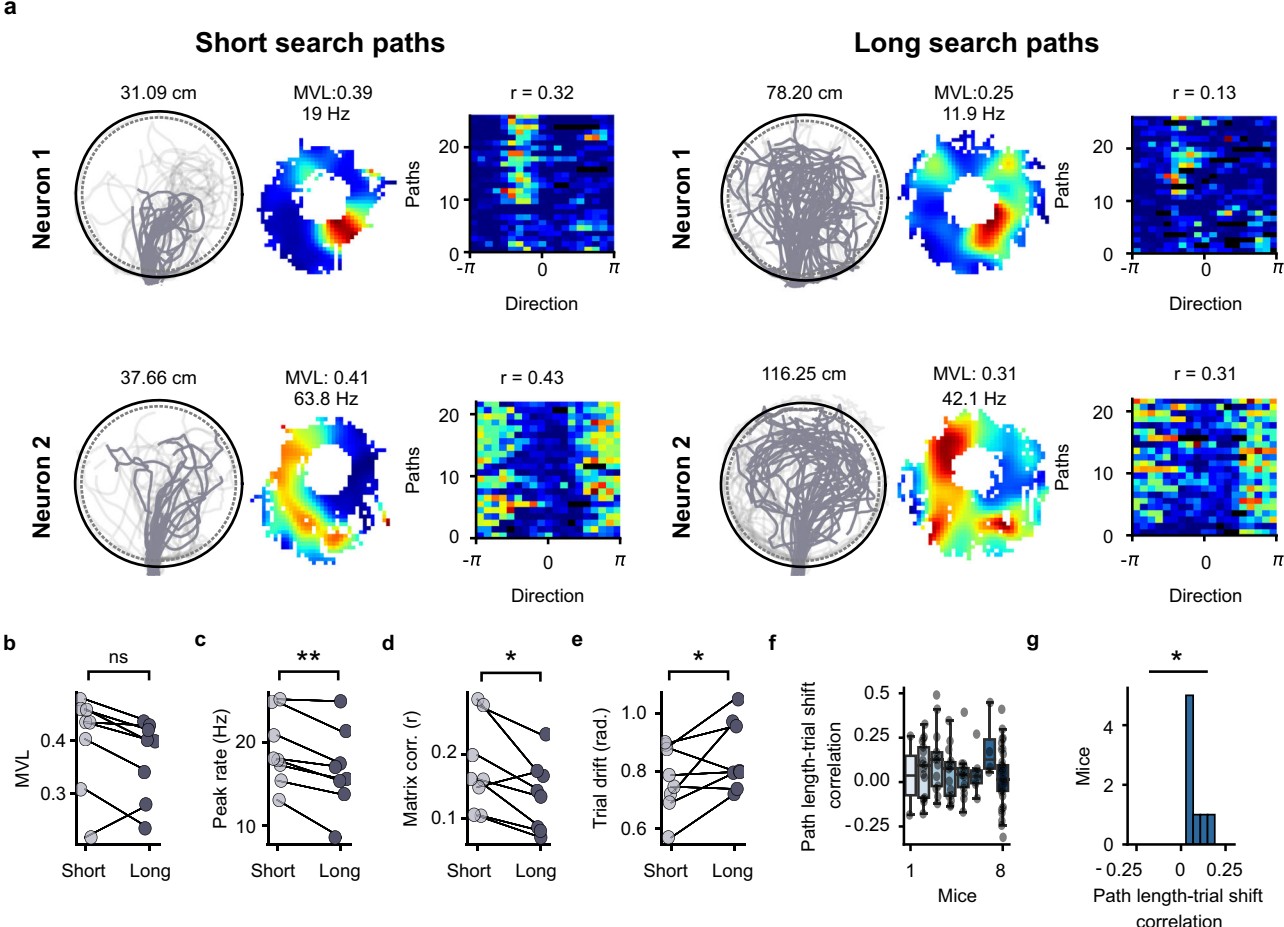

**Fig. 6 | Search path length negatively influences the directional selectivity of lever-anchored firing fields during dark trials. a** Examples of two neurons with a lever-box-anchored firing field during dark trials. Their firing activity around the lever box is shown separately for trials with short (left) and long (right) search paths. For each neuron, three plots are shown for dark trials with short and long search paths. Left: Search paths (dark gray) of the animal during dark trials. The number indicates the median length of the search path. Middle: 2D lever-box-centered firing rate map. The number indicates the peak firing rate. Right: Trial matrix of the neuron. Each row of the matrix is the firing rate as a function of the direction of the mouse from the lever-box center for a single trial. The number above the matrix refers to the trial matrix correlation. **b**–**e** Quantification of the differences in directional selectivity around the lever box between trials with short and long search paths. Only neurons with a lever-box-anchored field in the Cardinal reference frame were considered (n = 8 mice, 109 cells). **b** Mean vector length (MVL) of the firing rate of the neuron around the lever (n = 8 mice, one-sided Wilcoxon signed-rank test, $P = 7.8 \times 10^{-3}$). **c** Peak rate of the directional polar histogram (one-sided Wilcoxon signed-rank test, $P = 3.9 \times 10^{-3}$). **d** Trial matrix correlation (one-sided Wilcoxon signed-rank test, $P = 7.8 \times 10^{-3}$). **e** Directional trial drift (one-sided Wilcoxon signed-rank test, $P = 0.15$). **f** Distribution of Pearson correlation coefficients between the search path length and the trial drift for each mouse. Each point represents a lever-box-anchored firing field. **g** Distribution of correlation between the search path length and the trial drift (n = 8 mice, one-sided Wilcoxon signed-rank test, $P = 3.9 \times 10^{-3}$). *$P < 0.05$, **$P < 0.005$, ns non-significant. Source data are provided as a Source Data file.

individual trials and homing direction. We first calculated the homing direction as the angle between two vectors originating at the center of the lever box, one pointing towards the bridge and the other towards where the mouse reached the periphery of the arena. For each session, we identified the median homing direction and classified trials into two groups based on the homing direction being below or above the median (Fig. 7c i middle column). We calculated the polar firing rate histogram for dark trials with homing direction below or above the median homing direction (Fig. 7c i). Several neurons showed a change of preferred direction associated with homing direction. To determine whether these changes in the preferred direction were statistically significant, we calculated the changes after shuffling the homing direction vectors 500 times. The observed changes in the preferred direction associated with homing direction were statistically significant ($P < 0.05$) in 25% of the neurons (27/109 neurons). The change in preferred direction was also statistically significant when averaging the tuning curves of all neurons with lever-box-anchored fields (n = 109 neurons, Supplementary Fig. 10 a–d). Importantly, the change in firing

direction associated with homing direction was observed when limiting the analysis to trials where the lever was located near the center of the arena, ruling out the possibility that the change in firing direction was caused by different lever positions (Supplementary Fig. 10 e–h).

We also tested the hypothesis of a linear relationship between the trial directional drift in the firing direction of a neuron and the homing direction (Fig. 7c ii, iii). The relationship between trial firing direction and homing direction was visualized by plotting the trial matrix after sorting trials based on homing direction (Fig. 7c ii). We performed a circular-circular correlation between the trial drift and the homing direction of the mouse (Fig. 7c iii). The correlation was significant in 26.9 % of the neurons with lever-box-anchored fields. The r value was positive in most neurons with a significant correlation (24.1 vs. 2.8%, Fig. 7d). At the population level, we observed significantly more positive correlation coefficients (Fig. 7e), indicating that a clockwise trial drift in the activity of the neuron led to a clockwise rotation of the homing direction. We observed neurons with a significant positive correlation in 7 of the 8 mice in which

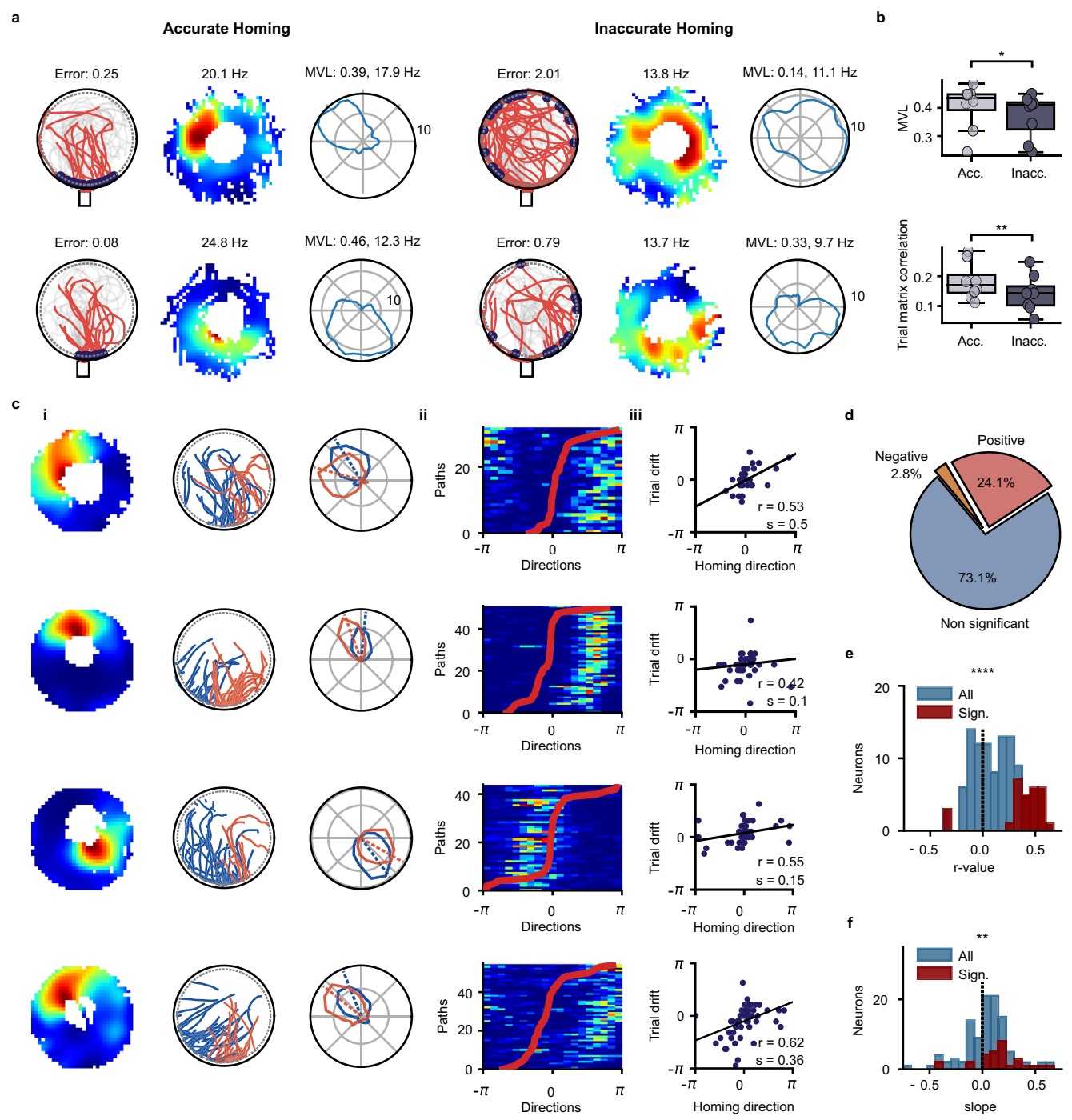

lever-box-anchored fields were recorded. The slopes of the regression line (as shown in Fig. 7c iii) were generally smaller than one (Fig. 7f), indicating that the trial drift in the activity of the neuron was generally smaller than the change in homing direction. These results demonstrate that the directional firing of hippocampal neurons around the lever was linked to the homing direction of the mouse.

## Discussion

There is strong experimental and theoretical evidence for the role of hippocampal and parahippocampal brain areas in navigation based on path integration[2,12–14,44–47]. Considerable progress has been made in understanding how the activity of place cells and other spatially selective neurons are updated via path integration[19,23–26,48–50]. However,

few studies considered the activity of spatially selective neurons in the context of path-integration-based navigation in freely moving animals[19]. This raises two questions: whether the firing patterns present in animals actively engaged in path-integration-based navigation are the same as during other types of navigation and whether the activity patterns of spatially selective neurons during path-integration-based navigation correlate with successful navigation. We developed a H-PI task and addressed these questions for hippocampal neurons.

On the AutoPI task, mice need to enter a circular arena, search for a lever box, press the lever, and then come back to their home box to receive a food reward. We found that cell ensembles active during the task differed from those active during random foraging. During the AutoPI task, different cell ensembles were active depending on whether visual landmarks were available and whether the animal was

**Fig. 7 | Direction of lever-box-anchored fields predicts homing direction during dark trials. a** Example of two neurons (one per row) with a lever-anchored firing field during dark trials. The activity at the lever box is shown separately for trials with accurate (left) and inaccurate (right) homing. Left: Homing paths of the mouse. The blue dots indicate the locations at which the mouse reached the arena periphery. The number indicates the median homing error in radian at the periphery of the arena. Middle: 2D lever-box-centered firing rate map. The number indicates the peak firing rate. Right: Directional firing rate histograms in the Cardinal reference frame. The mean vector length (MVL) and peak rate are shown above the plot. **b** The mean vector length of directional firing rate histograms (n = 8 mice, two-sided Wilcoxon signed-ranked test, P = 0.039) and trial matrix correlation (n = 8 mice, two-sided Wilcoxon signed-ranked test, P = 7.8 × 10⁻³) obtained from dark trials with accurate and inaccurate homing. **c** Example of four neurons with a significant correlation between the direction of their firing field around the lever and the homing direction of the mouse during dark trials. **i** Left: 2D lever-box-centered firing rate map. Middle: Homing paths of the mouse. The color of the homing path indicates whether the mouse reached the periphery of the arena on the left (blue) or right (red) side of the mean homing direction of the recording session. Right: Directional firing rate histogram for trials in which the mouse reached the periphery of the arena on the left (blue) or right (red) side of the mean homing direction. The dashed lines indicate the mean firing direction of the

directional firing rate histograms. Clockwise homing error are associated with a clockwise rotation of the lever-anchored fields, and vice versa. **ii** Trial matrix and homing direction of the mouse. The trial matrix shows the firing rate of the neuron on each dark trial as a function of the direction around the lever box. The trials were sorted according to the homing direction of the mouse, which is displayed as a red line. **iii** Trial drift of the neuron on each dark trial plotted against the homing direction of the mouse. The peak firing rate direction was rotated so that its preferred direction was 0. The regression line, together with the correlation coefficient (r) and slope of the regression line (s), are shown. The trial drift of the neurons predicted the homing direction of the mouse. **d** Percentage of the lever-anchored neurons with a significantly positive, significantly negative, or non-significant correlation between the trial drift of the neuron and the homing direction of the mouse. **e** Distribution of correlation coefficients between the trial drift of the neuron and the homing direction of the mouse for all neurons with a lever-anchored firing field (n = 109 cells, two-sided Wilcoxon signed-rank test, P = 4.05 × 10⁻⁸). **f** Distribution of regression slopes between the trial drift of the neurons and the homing direction of the mice for all neurons with a lever-box-anchored firing field (n = 109 cells, two-sided Wilcoxon signed-rank test, P = 0.0019). *P < 0.05, **P < 0.01, ****P < 0.0001. Source data are provided as a Source Data file.

engaged in search or homing behavior. Besides traditional place cells that fired at a fixed location on the arena, we observed several neurons that either encoded the distance between the animal and the lever box or fired when the animal was in close proximity to the lever box, independently of the lever position on the arena. The neurons with firing fields around the lever box were active only when the animal was at a given direction relative to the center of the lever box, providing directional information in a Cardinal reference frame. When homing performance depended on path integration (i.e., dark trials), the directional tuning of lever-anchored fields was reduced after long search paths and before inaccurate homing. The peak firing direction of neurons with lever-box-anchored fields on single trials predicted the homing direction of the mouse.

Several behavioral tasks exist to assess path integration in freely moving rodents[3,16,45,51]. The food-carrying task on a circular arena is the most common one[9]. The AutoPI task has several advantages over the food-carrying task. Firstly, mice can perform approximately 75 trials per hour on the AutoPI task and up to 150 trials per day. This is one order of magnitude more than with the food-carrying task, in which the number of trials is limited by the large size of the food reward on each trial[15,33]. In the current study, a higher number of trials was essential to characterize the firing patterns of hippocampal neurons and to study the relationship between neuronal activity and homing performance. A second advantage of the AutoPI task is that the task is fully automated. This reduces the variability between experiments and enables the testing of several animals in parallel. The experimenter also monitors the experiments from a different room, reducing the risk of interfering with the animal's behavior.

Distance estimation using path integration has also been assessed in head-fixed rodents in virtual reality[40] and on linear tracks in real-world navigation[51]. In the virtual reality paradigm, mice were trained to run in a VR linear track and stop at a reward location. The location of the reward zone was marked by local visual cues on 4 out of 5 trials. On every fifth trial, the local visual cues were absent, and the animal had to reach the reward zone by estimating its running distance. This paradigm has the advantages of being automated and assessing selectively linear path integration. In the AutoPI task, the animal moves freely in 2D space, ensuring a more natural coupling of motor movements and vestibular stimulation. In addition, high homing accuracy on the AutoPI task likely requires the integration of both linear and angular self-motion cues. Another distinction is that the homing response in the AutoPI task changes across trials, whereas the correct motor response remains the same across trials on the VR linear track. Thus, in the AutoPI task, mice must use within-trial information gathered

during the search path to calculate their homing path. It will be interesting to test whether different neuronal circuits are recruited when the appropriate motor response to obtain a reward remains the same or varies across trials.

We observed a strong reorganization of hippocampal ensembles between a random foraging session and the AutoPI task, even though both experiments took place on similar arenas located at the same position within the experimental room. These results are consistent with the previous observation that place fields become more directional and often change location or turn silent when an animal transitions from a random search task to more stereotyped search trajectories on an arena[31]. Thus, different behaviors trigger remapping in the hippocampus. Similarly, differences in navigational demands might be responsible for the remapping observed on the AutoPI task when visual landmarks were eliminated. In previous experiments, it was shown that eliminating visual landmarks during a random foraging task does not always lead to a significant reorganization of the majority of the place fields[21,52]. During random foraging, the navigational demands remain largely unchanged with and without visual landmarks, as the animal can perform a random walk without having a known destination for its ongoing movements. In the AutoPI task, removing visual landmarks implies that instead of running directly to a visual target, the mouse has to explore the arena without knowing the location at which the search behavior will end. This change in navigational demands is reflected in overt behavior changes like reduced running speed and increased search path lengths and complexity. We propose that the changes in navigational strategy contributed to the reorganization of hippocampal cell ensembles when access to visual landmarks was changed[42,43].

During search and homing behavior, we observed that hippocampal fields could encode the distance to or from the lever box. These fields were recorded together with fields that had a stable location relative to other room cues. During the search path, neurons encoding the distance to the lever box were observed only when the lever box was visible. During homing behavior, encoding of the distance from the lever box was observed during both light and dark trials. This indicates that the distance from the lever box could be estimated using path integration, as previously suggested based on the findings from linear track experiments[23,53]. We also found that approximately 25% of neurons fired in close proximity (<10 cm) of the lever box. These neurons fired in a lever-box-centered reference frame, being active on most trials independently of the lever-box position on the arena. Hippocampal neurons firing in a goal/landmark-centered reference frame had been

reported in rats navigating to moving landmarks on an arena[24]. In addition, during random foraging, hippocampal cells with fields anchored to a moving border have also been reported[54]. Our data extend these previous findings by showing that object-related firing fields do not require visual inputs, and that direction of the firing fields relative to a task-relevant object is encoded in a Cardinal reference frame. The directional component of the lever-box-anchored fields is similar to that of landmark-vector cells of the hippocampus and medial entorhinal cortex, which also fire at a specific angle from an object[55,56]. It seems likely that the directional component of the lever-box-anchored fields is directly or indirectly related to that of head-direction cells. During dark trials, the head-direction system can integrate angular self-motion cues to maintain an estimate of head direction, and their activity is also correlated with homing direction[19]. It is possible that the translation from head-direction activity to a code that reflects the direction relative to the lever box could take place upstream in the medial entorhinal cortex, which contains head-direction cells, object-vector cells, and border cells. Border cells have been shown to respond to the presence of walls with a specific orientation[57], and their directional selectivity remains coherent with the preferred direction of head-direction cells. Inputs from MEC border cells to the hippocampus[58] could contribute to the lever-box-anchored firing fields observed during the AutoPI task. Alternatively, object-vector cells could also provide distance- and direction-dependent inputs to neurons with lever-box-anchored fields. A better understanding of the mechanisms behind lever-box-anchored firing fields will require characterizing the activity patterns of neurons in the different layers of the entorhinal cortex during the AutoPI task.

Homing during dark trials of the AutoPI task requires the animal to integrate its linear and angular movements during the search path to determine in which direction to run to return to the home base. Extra care was taken to minimize the use of auditory and olfactory cues. For instance, white noise was played from above the arena to mask any uncontrolled auditory cues. The arena was also rotated between trials to prevent any odor traces on the arena to guide homing. We also found that homing performance in dark trials remained above chance level when we used a strong airflow above the arena to interfere with the putative use of odor gradients to locate the home base. Additional behavioral evidence for the use of path integration during dark trials comes from positive correlations between search path length and search path duration with homing error. These correlations are consistent with the idea that noise in the path integration process accumulates with the length of the search path[10,19,39,59].

The hippocampal representation of direction around the lever box appeared to be updated from a path integration process during dark trials. Indeed, we observed increased variability in the firing direction of lever-box-anchored neurons with longer search paths. One particularity of the lever-anchored firing fields is that the location of their firing field is set relative to the lever box and not the global reference frame of the recording room, which includes the starting point of the animal. Thus, errors in the integration of translational movements during the search path should not affect the stability of the firing field as the x,y reference frame of these fields is presumably set when the mouse encounters the lever box. Therefore, the variability in the lever-anchored firing fields reflects primarily errors in the integration of directional self-motion cues rather than both directional and translational information. It should be noted that although linear path integration is not hypothesized to be critical to set the directional firing fields around the lever, it is required for optimal homing on the AutoPI task.

Hippocampal lesions have been shown to impair homing behavior[12]. Here we found that approximately 25% of hippocampal pyramidal cells had directional firing fields around the lever. In a subset (24%) of these neurons with a lever-box-anchored field, the activity on single trials predicted the homing direction of the mouse. When the lever-box-anchored firing fields were rotated clockwise relative to their average direction, the homing direction of the mouse was also rotated clockwise. Although neurons firing near objects had been observed before in the hippocampus and medial entorhinal cortex[56,60], our results demonstrate that object-associated firing fields encode information that predicts homing behavior. Although no causality can be inferred from the current study, we hypothesize that these lever-box-anchored firing fields contribute to the homing behavior of the animal. Their exact contribution could be several-fold. Firstly, they could provide directional information within a world-centered cardinal reference frame that could help plan the homing direction. Secondly, as the animal runs around the lever box, a sequence of lever-box-anchored neurons is activated, which might represent this particular phase of every trial that immediately precedes choosing a homing direction[61]. Hippocampal sequences have been hypothesized to support associations between objects, events, and future actions[62–65]. In this context, hippocampal neurons might contribute to homing by representing not only spatial information but also the series of meaningful events and decision points that constitute a trial, thereby providing a cognitive plan to organize homing behavior. Studies in which the activity of hippocampal neurons is manipulated at different moments during a homing task will be needed to test whether the activity patterns observed in our study are causally linked to homing behavior.

We have shown that classical lever operant protocols can be extended to study homing behavior based on path integration. We observed significant remapping of hippocampal spatial representations in mice performing the AutoPI task compared to a random foraging task. Remapping was also observed between search and homing behavior and light and dark trials during the AutoPI task. We found that approximately 25% of hippocampal neurons fired when the animal ran around the lever box before initiating its homing behavior. These neurons fired in a reference frame centered on the variably placed lever box, with a fixed orientation relative to the recording room. The activity of the lever-box-anchored firing fields predicted the homing direction of the mice. A future goal will be to determine whether the spatially selective neurons located in the medial entorhinal cortex, which provide inputs to the hippocampus, have a global or multiple reference frames during homing based on path integration.

## Methods

### Apparatus

The AutoPI apparatus consisted of a home base (20 × 30 × 30.5 cm) and a circular arena (diameter: 80 cm) elevated 45 cm above the floor. The home base had an inverted sliding door on its front wall. The door could be lowered below the home base floor to give access to a bridge (10 × 10 cm) that extended towards the arena. A food magazine was attached to the back wall of the home base. The magazine was equipped with an infrared beam to detect the presence of the animal at the magazine. A pellet dispenser attached to the outside wall of the home base delivered food rewards (AIN-76A Rodent tablets 5 mg, TestDiet) in the magazine via a small plastic tube.

The circular arena was mounted on a tapered roller bearing to allow its rotation during the task. Eight wall inserts were fixed to the edge of the arena, creating 1.6 cm high walls along the edge of the arena. There was a 10 cm wide opening between the wall inserts. These openings were located at multiples of 45° on the arena edge, creating eight potential exit points to access the home-base bridge. The arena was always oriented to have one of the eight exit points aligned with the bridge.

The lever box was built on a 2-wheel-drive mobile platform (116 × 82 × 80 mm). The mobile platform contained two servo motors, an Arduino Nano connected to a radio frequency module (NRF24L01), and a 2000mAh lithium battery. The lever itself was 13 × 10 mm.

 

Pressing the lever broke an infrared beam located inside the lever box. The radio frequency module was used to establish bi-directional communication between the Arduino Nano of the lever box and a second Arduino attached to the microcomputer controlling the task (Jetson Xavier Nx or Raspberry Pi 4).

The arena rotation, movement of the door, and delivery of food rewards were operated via Arduino Uno microcontrollers, digital stepper motor drivers (Stepperonline, DM542T), and N17 stepper motors.

The behavior of the animal was monitored with two cameras, one above the circular arena and one camera above the home base. The cameras were connected to a microcomputer (Jetson Xavier Nx or Raspberry PI 4), and the videos were recorded at 30 Hz (640 × 480 pixels) for further offline processing.

The logic of the task was controlled by a Python script running on a microcomputer. We used the Robot Operating System (ROS, https://www.ros.org) as a framework to facilitate the communication between different programs (nodes) controlling individual parts of the system (door, arena, pellet dispenser, cameras, log). ROS allowed us to coordinate computer programs running on different computers, microcomputers, and Arduinos. During the task, a log file containing task-related events (lever press, food delivery, magazine IR beam break, door operation, arena rotation, etc.) with their respective timestamps was created.

The lever box was moved to a different location using a closed-loop navigation system. Before any movement, the current location and orientation of the lever box were estimated using DeepLabCut (https://github.com/DeepLabCut/DeepLabCut). The next lever position and orientation were randomly selected within a circle centered on the arena center with a radius of 75% of that of the arena. The movement vector between the current and future pose of the lever was calculated. The lever was instructed to move along this vector via radio frequency communication. The new position and orientation of the lever were then confirmed using DeepLabCut, and any substantial error was corrected.

**Lights and white noise.** Several infrared light sources remained turned on at all times during training and testing. LED stripes above the arena and the home base were the only sources of visible light around the setup. Visible lights could be turned on and off from an Arduino equipped with a relay module. A white noise (70 dB) was generated from a speaker located directly above or below the arena.

An electric fan (35 cm diameter) was located above the arena and could be switched on and off to create an airflow directed at the center of the arena. This fan was used only in a subset of experiments with six mice (Supplementary Fig. 5).

**Subjects**

The subjects were 3- to 6-month-old male wild-type C57BL/6 mice. They were singly housed in 26 × 20 × 14 cm high cages containing 2 cm of sawdust and 2–3 facial tissues. Mice were kept on a 12 h light-dark schedule with all procedures performed during the light phase. All experiments were carried out in accordance with the European Committees Directive (86/609/EEC) and were approved by the Governmental Supervisory Panel on Animal Experiments of Baden Württemberg in Karlsruhe (G-236/20). Before starting familiarization to the home base, each mouse was handled by the experimenter for 10 min per day for three days. Thirteen male mice were included in the behavioral experiments without electrophysiological recordings, and nine were used in the electrophysiological experiments. An additional six mice were included to test the effect of an artificial airflow above the arena on homing accuracy.

**Food restriction.** Three days before the familiarization to the home base started, mice were put on a diet to reduce their weight to

approximately 85% of their normal weight. Mice were weighed and fed once a day towards the end of the light phase. Water was available throughout the day. The food restriction continued until the end of the experiment.

**Training procedure**

**Familiarization with the home base.** The animal was placed in the home base for 20 min per day for three days. The home base door remained closed, and a food pellet was delivered in the food magazine every 30 s. The lever was not in the home base during familiarization. Familiarization took place under normal illumination.

**Lever training.** Lever training started after the familiarization. Each daily lever training session stopped after 30 min or after the mouse received 100 rewards, whichever occurred first. The lever box was placed at one of six potential positions at the beginning of the session and remained there for the entire session (Supplementary Fig. 1). A lever press led to a food reward being delivered in the food magazine. To obtain the next reward, the mouse had to break the infrared beam of the food magazine between lever presses. This procedure ensured that the mouse learned to visit the food magazine after pressing the lever. The lever box position in the home base changed between days.

After six days of ever training inside the home base, the sliding door was opened and the lever box was placed on the bridge. The lever was placed at the beginning of the bridge for two days and moved to the end of the bridge for two days. Once the mouse readily pressed the lever at the end of the bridge, the lever box was placed on the arena, approximately 10 cm from the bridge. The lever was progressively moved towards the center of the arena after training sessions in which the mouse received at least 70 rewards. The lever training procedure took approximately 16–20 days.

**Arena rotations and door operations.** Once the animal pressed the lever located at the center of the arena, we introduced door movement between lever presses. The sliding door opened at the beginning of a trial and closed when the mouse broke the infrared beam of the food magazine following a lever press. After two days, arena rotations were performed when the mouse was confined to the home base. For two sessions, the lever was at the center of the arena, and the arena angle varied between -45° to 45°. The range was then increased to -90° to 90° for two sessions, and finally, all eight possible orientations were used. During this phase, the mice learned to press the lever independently of its orientation. White noise (approximately 65 dB) was also introduced at that point and played throughout the sessions.

**Lever movement.** During this final training step, the arena was rotated between every trial to one of the eight possible orientations, and the lever was moved to a random location every four trials. This procedure ensured the mouse could press the lever independently of its orientation and position. The mouse was always confined to the home base during arena rotations and lever movements. Mice were trained in this protocol for five days. All training steps described so far occurred with the visible light sources turned on.

**AutoPI task**

Once training was completed, the AutoPI protocol started. For experiments without electrophysiological recordings, a session ended when the mouse completed 100 trials or 60 min elapsed, whichever came first. When electrophysiological recordings were performed, up to 150 trials were performed over a duration of approximately 90 min.

A trial started when the home base door opened. The mouse then left the home base, searched for and pressed the lever on the arena. A lever press triggered the delivery of a food reward in the home based food magazine. The mouse then returned to the home base to collect the food reward. The trial ended when the mouse broke the infrared

beam of the food magazine, causing the home base door to close. If the mouse returned to the food magazine without having pressed the lever, the trial continued, allowing the mouse to perform several journeys on the arena within a single trial.

Every session started with seven trials with visible light (light trials). Thereafter, the visible light sources were turned on or off at the beginning of each trial, creating a sequence alternating between light and dark trials. White noise was played throughout the session.

If the lever was not pressed within 240 s after the beginning of a trial, the visible lights were turned on (in the case of dark trials) and the trial ended at the next food magazine infrared beam break.

After every trial, the arena was rotated to one of the eight possible orientations (multiple of 45°). The lever was moved to a new random position and orientation every 4th trial. Note that the lever position also changed between the other three trials because of the arena rotation. This ensured that the search and homing paths were different across trials.

During the AutoPI task, the experimenter monitored the behavior of the mouse and hardware from an adjacent room.

### Artificial airflow above the arena

To test whether the odor cues originating from the bridge and home base contribute to the homing behavior of the mice, we performed a control experiment in which we interfered with odor gradients on the arena using a strong airflow. We fixed a fan (35 cm diameter) 1.85 m above the arena and directed the airflow toward the center of the arena. The positive air pressure above the arena should prevent the use of odor gradients to guide homing behavior. We tested six mice in this experiment over three days. Mice performed 35 trials with no airflow, followed by 35 trials with airflow, and up to 30 more trials without airflow (ABA protocol).

### Analysis of behavioral data

The data analysis was performed in Python using the following packages: NumPy, Pandas, Scipy, Scikit-Learn, Statsmodels, OpenCV, Deeplabcut, Matplotlib, and Seaborn.

**Object detection from the video.** The arena center and arena radius in the video were calculated using the *houghCircles* function from the OpenCV library. We trained convolutional neural networks (DeepLabCut) to detect the position of the bridge, lever, and mouse in the videos[66]. The position of the mouse in each video frame was extracted offline. Four body parts were tracked: the nose, two ears, and the base of the tail. The midpoint between the ears was used as the animal position.

**Trial segmentation.** The beginning and end times of each trial were obtained from the door opening and closing times in the event log file. A trial was divided into journeys, which started each time the mouse ran from the bridge to the arena. If the mouse pressed the lever during a journey, the mouse's running path was divided into search and homing paths. The search path ranged from the beginning of the journey until the animal reached the lever. The mouse was considered to be at the lever when it was less than 10 cm from the wall of the lever box. The homing path ranged from when the mouse left the lever until it reached the periphery of the arena. We considered that the mouse reached the arena periphery if it was less than 3 cm from the edge of the arena. Trials in which the center of the lever box was at a distance larger than 30 cm from the center of the arena were not used in the analysis.

For each search and homing path, we calculated its length, duration, average speed, and complexity. Path complexity was defined as $\ln((0 - MVL + 1)*100)$, where MVL was the mean vector length of the movement direction vector within each path. Paths containing several changes of direction had a high complexity score. We applied a logarithmic transformation to this score to reduce the skewness of the distribution.

### Surgical procedure

Nine mice were implanted with one, two, or four Buzsaki32 silicon probes (NeuroNexus) aimed at the CA1 region of the dorsal hippocampus. The probes were mounted on microdrives that allowed independent movement in the dorsoventral axis. Mice were anesthetized with isoflurane (1–3%) and fixed to the stereotaxic instrument. The skull was exposed, and two miniature screws were inserted into the skull. One screw located above the cerebellum served as ground electrodes. The skull and dura above the hippocampus were removed and the probes were implanted at the following coordinates (ML: ±1.8 mm from the midline, AP: 2.0 mm posterior to bregma). The probe tips were advanced 0.6 mm into the cortex and the microdrives were fixed to the skull with dental cement. During the first 72 h post-surgery, mice received a s.c. injection of Carprofen (0.1 mg/kg; Rymadil) every 8 h. Mice were given a week to recover after surgery. The probes were moved down by approximately 50 μm after each recording session.

### Electrophysiological recordings, spike extraction, and spike clustering

Mice were connected to the data acquisition system (RHD2000-Series Amplifier Evaluation System, Intan Technologies, analog bandwidth 0.09–7603.77 Hz, sampling rate 20 kHz) via a lightweight cable. The recording was controlled using ktan software (https://github.com/kevin-allen/ktan). Kilosort2 (https://github.com/jamesjun/Kilosort2) was used for spike extraction and clustering. Automatically generated clusters were visually inspected and manually refined with Phy (https://github.com/cortex-lab/phy).

The quality of spike clusters was estimated from the spike-time autocorrelation. A refractory period ratio was calculated from the spike-time autocorrelation from 0 to 25 ms (bin size: 0.5). The mean number of spikes from 0 to 1.5 ms was divided by the maximum number of spikes in any bin between 5 and 25 ms. Any cluster with a refractory period ratio larger than 0.25 was discarded.

In experiments with electrophysiological recordings, the position of the mouse was estimated from the position of infrared-LEDs (wavelength 940 nm) located on both sides of the head of the mouse. Three LEDs were used on one side and one on the other. The distance between the two LED groups was 3 cm. Two video cameras fitted with long-pass filters (Cut-On wavelength: 800 nm) and located directly above the recording environment monitored the position of the LEDs at 30 or 50 Hz. The location of the mouse was extracted online from the position of the LEDs (https://github.com/kevin-allen/positrack2).

**Cable actuator.** To minimize any interference of the recording cable with the behavior of the mice, we developed a 2D motorized linear actuator to keep the recording cable directly above the head of the mouse. The cable actuator was located 153 cm above the arena and had a movement range of 85 × 130 cm in the horizontal plane. The recording cable was attached to the cable actuator. The aim of the actuator was that its cable attachment remained directly above the mouse. This was achieved by calculating the speed of the mouse in the x-axis and y-axis from the online position tracking system and setting the speed of the actuator to the same values. The movement of the actuator was controlled by an Arduino Uno microcontroller, digital stepper motor drivers (Stepperonline, DM542T), and three N17 stepper motors.

### Analysis of electrophysiological data
**Classification of putative pyramidal cells and interneurons.** The presence of ripples (120–200 Hz) during rest trials was used to determine whether the recording sites of each shank of the silicon

probe were located in the CA1 pyramidal cell layer. Only neurons recorded from shanks with clear ripples were considered in the analysis. We then used the mean firing rates, the spike-time autocorrelations, and the mean waveforms of the neurons to classify the neurons as pyramidal cells or interneurons (Supplementary Fig. 6). Before clustering, we applied principal component analysis to the mean waveforms and spike-time autocorrelations and retained their respective first three principal components. We used the resulting six features together with the mean firing rate of the neurons (total of 7 features) as inputs for a K-means clustering algorithm with k = 2. The neurons of the cluster with the lowest firing rate were considered pyramidal cells.

**Firing rate maps.** Firing rate maps were generated by dividing the circular platform into 3 × 3 cm bins. The time in seconds spent in each bin was calculated, and this occupancy map was smoothed with a Gaussian kernel (standard deviation of 5 cm). The number of spikes in each bin was divided by the smoothed occupancy map to obtain a firing rate map. A smoothing kernel (standard deviation of 5 cm) was applied to the firing rate map. For firing rate maps limited to the area between the center of the arena and the bridge (as in Fig. 3e), the bin size was either 2.5 × 2.5 cm (Figs. 3) or 5 × 5 cm (Supplementary Fig. 7). Firing rate map similarity was the correlation coefficient between the firing rate maps of two neurons in the same condition.

**Firing rate histograms.** Firing rate histograms of search and homing paths were calculated by merging all paths of the same condition from one of the four possible combinations of trial types (light and dark) and behaviors (search and homing): Search-Light, Search-Dark, Homing-Light, and Homing-Dark. Only paths from journeys including a lever press were considered. For each video sample within a path, we calculated the y-axis coordinate of the animal position (axis parallel to a vector with origin at the center of the bridge and pointing towards the center of the arena) and the animal distance to the lever. For every neuron, we created a firing rate histogram for the y-axis coordinate and the distance from the lever box in each of the four task conditions (Search-Light, Search-Dark, Homing-Light, Homing-Dark) (8 histograms per neuron).

All histograms contained 20 bins. The time in seconds spent in each bin was calculated, and this occupancy histogram was smoothed with a Gaussian kernel (standard deviation of 1 bin). The number of spikes in each bin was divided by the smoothed occupancy histogram to obtain a firing rate map. A smoothing kernel (standard deviation of 1 bin) was applied to the firing rate histogram.

To allow direct comparison between the firing rate histograms of light and dark conditions, the minimum and maximum values of the histogram were set to be the same across light and dark conditions. The maximal value was obtained by calculating the 90th percentile for a behavioral variable for light and dark trials separately and selecting the smallest 90th percentile of the two. The minimal value was -42.5 cm for the histograms with the y-axis coordinates. -42.5 cm corresponds to the empty gap between the arena and the bridge. The minimal value was 12.0 cm for the histograms with the distance from the lever box. This procedure to establish the range covered by the firing rate histograms ensured sufficient sampling in all bins.

Information scores of firing rate histograms was calculated as previously described[67], $\sum_{i=1}^{N} p_i \frac{\lambda_i}{\lambda} log_2 \frac{\lambda_i}{\lambda}$, where $p_i$ is the probability to be in bin $i$, $\lambda_i$ is the firing rate in bin $i$, and $\lambda$ is the mean firing rate of the neuron. No smoothing was applied to the histograms when calculating information scores.

**Trial firing rate matrices.** Trial firing rate matrices were created to perform analysis based on single paths on the arena. Each matrix represents the firing activity of one neuron. The values in the matrix are the firing rate of the neuron. Each row of the matrix represents a single trial and was usually limited to a specific part of the trial (e.g., search path, homing path, etc.). The x-axis represented one of the different behavioral variables, such as the y-axis coordinate of the animal position, the animal distance to the lever, or the direction of the mouse relative to the center of the lever box. The size of the bins along the x-axis of the matrix depended on the behavioral variables: y-axis coordinate of the animal position: 2 cm, animal distance from the lever: 2 cm, and direction of the mouse relative to the center of the lever box: 20°. Each row of the trial matrices was smoothed with a 1D Gaussian kernel with a standard deviation of 2 bins.

To quantify the reliability of the firing rate of a neuron between trials, we calculated a correlation matrix for all possible pairs of single-trial firing rate vectors (matrix rows) within one session (Supplementary Fig. 6). The mean of the correlation matrix, excluding the unit diagonal, was termed trial matrix correlation and reflected how similar the single-trial firing rate vectors were.

**Analysis of cell activity around the lever box.** The distance from the lever was defined as the distance between the center of the animal's head to the closest wall of the lever box. To analyze the firing rate of the neurons around the variably located lever box, we transformed the animal position into a lever-centered coordinate system by subtracting the coordinates of the lever box from the position of the mouse. Each position of the mouse can be seen as a 2D vector originating at 0,0 and pointing towards the direction of the mouse around the lever. Data from when the animal was more than 12 cm away from the lever box were excluded from the analysis.

We considered three possible directional reference frames in which the neurons could encode directional information (Fig. 5b). In the Cardinal reference frame, the direction of the animal around the lever box was measured from a vector pointing south. The original lever-centered data were in the Cardinal reference frame. In the Bridge reference frame, the direction of the mouse was measured from a vector originating at the center of the lever box (0,0) and pointing toward the center of the bridge. To transform the data from the Cardinal to the Bridge reference frame, we calculated the angle of the vector originating at the center of the lever box (0,0) and pointing toward the bridge relative to a vector pointing south (0,-1). We then rotated the Cardinal lever-centered mouse position data by this angle. In the Lever reference frame, the direction of the mouse is measured from a vector originating at the lever box center and pointing towards the lever. To transform the lever-centered data from the Cardinal to the Lever reference frame, we calculated the angle of the vector originating at the center of the lever box (0,0) and pointing towards the lever relative to a vector pointing south. We then rotated the Cardinal lever-centered mouse position data by this angle.

We generated 2D firing rate maps using the lever-centered data of the three directional reference frames. These maps were calculated like the standard 2D firing rate maps, but the bin size was 1 cm, and the SD of the smoothing kernel was 2 cm. To quantify the degree of directional selectivity of the neurons around the lever, we also calculated directional firing rate histograms. The directions around the lever were divided into 10° bins. The time in seconds spent in each bin was calculated, and this occupancy histogram was smoothed with a Gaussian kernel (standard deviation of 1 bin). The number of spikes in each bin was divided by the smoothed occupancy histogram to obtain a firing rate histogram. A smoothing kernel (standard deviation of 1 bin) was applied to the firing rate histogram. We used the peak firing rate and mean vector length of the directional firing rate histogram as a measure of directional selectivity.

**Directional trial drift.** To measure single-trial variability in the orientation of lever-anchored firing fields, we developed a measure called

the trial drift (Supplementary Fig. 9) calculated from the trial matrix. The x-axis of the trial matrix represented the direction of the mouse relative to the lever box. The trial matrix was first rolled in the x-axis until the maximal mean firing rate of the neuron was at direction 0 (Supplementary Fig. 9b). We then calculated an idealized tuning curve for the neurons by re-aligning all trials individually so that their peak firing rates were at direction 0 (Supplementary Fig. 9c). We next performed a crosscorrelation between the idealized tuning curve and each row of the trial matrix with the maximal mean firing rate at direction 0 (Supplementary Fig. 9d). The trial drift was the location of the peak in each row of the convoluted trial matrix (Supplementary Fig. 9e).

## Statistical analysis

The statistical significance was tested with the Mann-Whitney rank tests, ANOVAs, chi-square tests, and Wilcoxon signed-rank tests. The SciPy and statsmodels Python modules were used to perform these tests. When appropriate, we used a shuffling procedure to establish significance levels. The linear SVM model was implemented with Scikit-learn. When aggregating trial data to obtain one score per mouse, the mean of the trial scores was used.

**Box plot definitions**. All box plots indicate the median, interquartile range, and minimum to maximum values. The percentage range for whiskers (the lines extending from the box) is set to 1.5 times the interquartile range above the third quartile and below the first quartile.

## Reporting summary

Further information on research design is available in the Nature Portfolio Reporting Summary linked to this article.

## Data availability

The data generated in this study have been published on the Dryad platform under accession code https://doi.org/10.5061/dryad.crjdfn39x. Data frames used to generate the figures are provided in Supplementary Information/Source Data file. Source data are provided with this paper.

## Code availability

The code used in this study can be found in a Github repository https://github.com/Mrymna/Jazi_et.al_2023_noInt or via https://doi.org/10.5281/zenodo.8356898[68,69].

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

## Acknowledgements

This work was made possible by the Deutsche Forschungsgemeinschaft via Individual Research Grants (AL 1730/3-1 and AL 1730/4-1 K.A.) and Collaborative Research Centre (SFB-1436, Project-ID 425899996, K.A.). We also thank the Lautenschläger Foundation for supporting the Department of Clinical Neurobiology. We thank Marcel Weinreich for his help with developing the behavioral apparatus and Pascal Klein for testing the code uploaded to the GitHub repository.

## Author contributions

M.N.J. and K.A. developed the project; A.T. and K.A. created the source code for controlling the behavioral task. M.N.J., F.J.K., T.Y.Y., and K.A. designed, improved, and built the behavioral setup. M.N.J., M.S., B.B., S.R.C., C.G.V., and T.Y.Y trained mice. M.N.J recorded from CA1. Data analysis was performed by M.N.J and K.A. Writing was done by K.A. and M.N.J. Supervision and coordination of the project was done by K.A. Surgeries were done by K.A. and M.N.J.; M.N.J. and K.A. generated figures of the manuscript. The revision was done by M.N.J and K.A. Manuscript editing and proofreading were performed by all authors.

## Funding

## Competing interests

The authors declare no competing interests.
