## [Peer Review File · Nature Communications]

Hippocampal firing fields anchored to a moving object predict homing direction during path-integration-based behaviorREVIEWER COMMENTS

Reviewer #1 (Remarks to the Author):

This study applies a new behavioural task to investigate path integration mechanisms during homing behaviours. By recording from the hippocampus during this task, the authors show that hippocampal neural activity remaps according to the phase of the task and identify neurons with activity anchored to the lever box. The firing of these neurons in the dark appears to predict an animals homing direction, which is consistent with their reflecting the output of a path integration process.

The new task is impressive and the results are important. First, the study provides new evidence to challenge the general idea of the hippocampus as providing a stable world-centred reference frame, suggesting instead that representations are specific to the task at hand. Second, the study provides evidence for activity of some hippocampal neurons being driven by a path integration mechanism. While the manuscript is clearly written, the key results and their importance could be emphasised more clearly, particularly in the abstract.

1. The abstract and introduction could perhaps be reworded to make clearer what the big picture advance is.

In addition, I appreciate the intended focus of the study is on path integration mechanisms, but the remapping results are also of interest. I suggest to consider emphasizing further in the abstract and discussion. Similarly, the introduction might motivate the 'standard model' which predicts stable hippocampal place representations across all phases of the task and alternative models in which hippocampal representations are task-dependent.

2. Could mice use odour cues to return to the bridge? As I understand it, the arena is rotated after each trial to prevent the mice following a previous odour trail, but does it remain possible that they could orient using olfactory cues originating from the home base, or from fresh trails laid down during the search phase? This merits discussion.

3. It is good that Extended data Figures 3 and 4 show analyses using the mouse as a unit of analysis. For Figure 3H-K, I think the analysis in the extended data is more correct and should be used in the main figure. For Figure 4, the goal of showing the data distribution at the level of trials is appropriate, but the statistical analysis should still account for dependencies at the level of animals in order to avoid

overinflated degrees of freedom (and artificially low p-values). This could be done by adopting a hierarchical approach or by using the mice as unit of analysis (as in the extended data). It is also important for these analyses to report the test statistic and degrees of freedom alongside the p value.

4. Were probes moved between recording days? Are data reported from the same mouse for multiple days? If so, how is double counting of the same neuron avoided?

5. Over-inflation of the degrees of freedom for statistical analyses of the neuronal firing is also a potential concern. As presented the analyses implicitly assume that all neurons (or neuron pairs) are independent. This ignores dependencies that arise from neurons being recorded from the same animal and in the same session. I don't think the conclusions are likely to change in this case but it is nevertheless important to assess whether such dependencies exist and if necessary modify the analyses accordingly (see e.g. <https://www.jneurosci.org/content/38/26/5837>).

6. Could weak firing rate correlation between foraging and AutoPI conditions be caused by dark trials? What if these are excluded?

7. Firing rate similarity result. What was the correlation between first and second half of the foraging session?

8. I wonder if Figure 4a would be more useful if it showed the same neuron across all four conditions? As I understand it, at the moment it shows separate example neurons for each condition, but this leaves open the question of what each neuron's activity looks like in the other condition.

9. In testing for whether directional selectivity is influenced by search path length, it would be helpful to show the scores reported in Figure 6b-e as a function of path length (rather than differences). This could be considered first at the level of individual animals. According to the path integration hypothesis these scores should correlate with path length, at least for animals with a reasonable distribution of path lengths. Is this the case?

Related to this, for the analysis in Figure 6, is there a large range of path lengths for all animals? If not, then I wonder if the analysis is obscuring a larger true effect; if some animals have little variation in their path lengths then their difference scores (Figure 6b-e) would on average be close to zero.

10. Similar comments for Figure 7b as for Figure 6b-e.

11. Extended Data Figure 7. It could be helpful to briefly make clear the rationale for applying the cross correlation rather than simply using the peaks of the shifted trials (b).

12. The use of 'several neurons' when summarising the work is a little vague (e.g. abstract and discussion). Perhaps indicate the proportion?

Reviewer #2 (Remarks to the Author):

This paper reports a study in which hippocampal place cells were recorded from mice performing a continuous homing task, in which they had to search an area for a release lever and then navigate to the home box to retrieve the food made available by the lever. These trials were recorded both in light, when the animals could see the home box, and in darkness when they had to navigate using their internal sense of movement and space (i.e., using path integration). The study addressed two questions: (i) whether the firing patterns present in animals actively engaged in path-integration-based navigation are the same as during other types of navigation, and (ii) whether the activity patterns of spatially selective neurons during path-integration-based navigation are linked to successful navigation. The main findings are (i) that the place cells remapped their firing fields between phase of the task (search vs. homing) and between light and dark; (ii) that around 25% place cells had firing fields that were located in a given direction from the mobile lever at a fixed distance from it, and (iii) that parameters of the firing such as the field's spatial stability and direction relative to the lever correlated with navigation behavior. It is concluded that "the changes in navigational strategy were responsible for the reorganization of hippocampal cell ensembles

when access to visual landmarks was changed" and (in relation to the lever-anchored cells) "This observation rules out the simpler hypothesis that all hippocampal neurons fire in a global and stable reference frame that includes the starting point of their journey."

This is a nice behavioral task addressing an issue that has long been of interest to those studying spatial cognition, which is the neural mechanisms that underpin path integration. The data are well analyzed and nicely presented and the manuscript is well-written. However I have some concerns about whether (a) the conclusions are fully supported by the data, and (b) the extent to which the findings are novel. These are as follows.

The first major conclusion is that there is remapping between the search phase and the homing phase of the task. This is supported by the data illustrated in Fig. 3 where correlations are made between cell-pairs, with the decline in correlation across task phases used to support the notion that the cells

remapped. However this analysis is confounded by the change in behavior between these task phases, such that it is difficult to compare place fields since their sampling may not have been equivalent. For example, a place cell whose field was traversed during search but not homing would change its correlated firing in the two phases, even if its field remained the same. Thus, I am not convinced that remapping is convincingly shown here (the single illustrative examples are also not so convincing, and also highlight the uneven sampling). A better way to do this would be to match trajectories run for run between the two conditions, and ideally to use a many-cells correlation (population vector) rather than restricting to cell-pairs. In any case, even if there is remapping, such changes have been shown before in related situations (e.g., Markus et al who saw remapping between foraging and goal-directed search, and Geva-Sagiv et al who saw it between visual and echo-location navigation). Therefore it isn't clear that such a finding would be informative about path integration per se, so much as context processing more generally.

The second main question is whether the firing patterns of the cells are "linked" to successful navigation. The implication is made that such a linkage would imply a causal connection in which the cells drive the behavior. For example, in the Discussion they say "our results demonstrate that object associated firing fields encode information relevant for navigation. We hypothesize that these lever-box-anchored firing fields contribute to the homing behavior of the animal". However they acknowledge themselves immediately afterwards that this remains speculative – it could be that the behaviour informs the fields, or that another factor influences both behaviour and field location (the head direction signal being an obvious candidate). Thus, the results do not answer the second question either.

The third main finding is of cells that anchor their firing to the mobile lever rather than to the boundaries of the environment. This is an interesting finding but not novel: The phenomenon of place cells attaching to mobile objects at a characteristic direction has been reported previously by Gothard et al., Rivard et al and Deshmukh and Knierim.

Overall, then, I feel that some more analysis is needed, the conclusions need to be modified, and there needs to be stronger linkage to previous work and the fact that the present work yields more of a replication than completely novel findings (which personally I find a strength of the paper, given the current focus on reproducible science).

Other comments:

Abstract: "These results demonstrate how neurons with object-anchored firing fields contribute to navigation." Is a little too vague to be a useful conclusion, and also makes the incorrect assumption discussed above about the causal link in these phenomena – these fields do not necessarily contribute to navigation (although they might).

L64+ “Among spatially selective neurons, hippocampal place cells are likely to contribute to H-PI. Indeed, hippocampal lesions affect H-PI, and place cells can remain spatially selective when external landmarks are removed... implying that a path integration process can control place cell activity” There is a similar causal direction issue here. The first statement is about how hippocampal cells could affect PI, and it doesn’t follow therefore that PI affects the cells.

L418 – this section began with extensive discussion of the distance between the firing of the cell and the lever but left out the other obvious point that this distance was at a fixed direction – the cell thus fired with a vectorial relationship to the lever. It is very confusing to have distance and direction separated out like this and I suggest to combine and just make the point about the vector. These are clearly “object-vector cells” and since this terminology already exists it would be best to use it for clarity.

Place field analysis – we need to see at least some of the spike plots as the heat maps are too uninformative as to what is going on. The examples in Fig. 5A are great and it would be good to see more like this. We also need some illustration of cluster separation quality. One suggestion is to have one clear example of the main phenomenon where we see the cluster, waveform, and spikeplot so as to gain a really clear picture of what the underlying data look like.

Was there any evidence of the prevalence of lever-anchored fields increasing or decreasing across days? (Or both, in different cells perhaps)

The overall conclusion doesn’t fully sum up the findings. There were two initial questions and the findings yielded a third observation, so all three of these should be summarized.

Statistics – The statistics are calculated on a per-cell basis, which is OK for basic physiological properties but not for more cognitively relevant ones as these are, where characteristics of the whole animal (such as its head direction orientation) come into play and introduce dependencies in the data. The statistics therefore need also to be calculated on a per-animal basis. I personally also prefer to see that statistics in the text itself rather than hidden away in a supplementary data table (although I understand the decluttering motivation).

Figures – a general comment: it would be good to have the take-home message of each plot included in the legend. These analyses are very hard to unpick and casual readers will not want to spend the time.

Also - it would be good to see the individual data points plotted on the bar charts

Fig. 2a – typo – missing g on “homing”

Fig. 4a – I didn't understand the point being made by this figure. Are the same three neurons being shown in every panel? (I assume so). The figure aims to show that lever-box location modulates place cell activity but according to my understanding only the third cell shows this. I think it would help to make this more explicit: show just two cells, one modulated by distance to the start box and one by distance to the lever box, and label these clearly as such. As mentioned above I think it would also be helpful to have the spike plots so as to get a clearer picture of what is actually happening. I think naïve readers also need some help in understanding the heat plots; the point being that a start-box-anchored cell will have fixed location in the left-hand plot and variable in the right, whereas a lever-anchored cell will show the reverse. Stepping back from this a bit, I don't find the linearized heat plots very intuitive and it might be easier to understand if the data were plotted as field centroids on a 2D plot, one with the origin centered on the start box and the other with it on the lever box.

Fig. 5a I found the colors of the spike plots hard to distinguish (albeit tasteful!). The take-home message of this figure seems to be the same as the previous –these could be distinguished more clearly (or else combined). In Fig. 5b it was not immediately obvious what the lever reference frame means because it hasn't been highlighted that the lever itself is rotationally asymmetric. (in fact it remains unclear how asymmetric it actually is – would a mouse even notice this?)

We thank the reviewers for providing constructive feedback that helped us improve our manuscript. We here provide a point-by-point response to the reviewers' comments.

Point-by-point response to the reviewers

Reviewer #1:

This study applies a new behavioural task to investigate path integration mechanisms during homing behaviours. By recording from the hippocampus during this task, the authors show that hippocampal neural activity remaps according to the phase of the task and identify neurons with activity anchored to the lever box. The firing of these neurons in the dark appears to predict an animals homing direction, which is consistent with their reflecting the output of a path integration process.

The new task is impressive and the results are important. First, the study provides new evidence to challenge the general idea of the hippocampus as providing a stable world-centred reference frame, suggesting instead that representations are specific to the task at hand. Second, the study provides evidence for activity of some hippocampal neurons being driven by a path integration mechanism. While the manuscript is clearly written, the key results and their importance could be emphasised more clearly, particularly in the abstract.

1. The abstract and introduction could perhaps be reworded to make clearer what the big picture advance is. In addition, I appreciate the intended focus of the study is on path integration mechanisms, but the remapping results are also of interest. I suggest to consider emphasizing further in the abstract and discussion. Similarly, the introduction might motivate the 'standard model' which predicts stable hippocampal place representations across all phases of the task and alternative models in which hippocampal representations are task-dependent.

We have modified the abstract to state that “hippocampal neurons remap between random foraging and AutoPI task, between trials in light and dark conditions, and between search and homing behavior.” We have also extended the Introduction section contrasting the “single world-centered reference frame” and “multiple reference frames” scenarios. The remapping is discussed in the section “Task-driven change in active cell ensembles” of the Discussion.

2. Could mice use odour cues to return to the bridge? As I understand it, the arena is rotated after each trial to prevent the mice following a previous odour trail, but does it remain possible that they could orient using olfactory cues originating from the home base, or from fresh trails laid down during the search phase? This merits discussion.

We agree with Reviewer 1 that this is an important issue. We performed additional experiments in 6 mice in which we introduced a strong airflow directed at the center of the arena using an electrical fan located above the arena. This airflow creates high air pressure at the center of the arena, making it unlikely that odor from around the arena could be used to return to the home base. Six mice were tested in sessions containing 35 trials with airflow and 65 trials without airflow (ABA protocol). Each mouse was tested for three daily sessions. Homing error with and without airflow was not significantly different (Fig. 1R).

Fig. 1R. Effect of an artificial airflow over the arena on the search and homing behavior. Error at periphery and running speed during search and homing for light (top row) and dark (bottom row) trials with or without airflow (On: Airflow present; Off: No airflow). There was no significant effect of airflow on homing error or running speed ($n = 6$ mice, Wilcoxon signed-ranked test, Light trials: Error at periphery, $P = 0.6$; Search speed, $P = 0.12$; Homing speed, $P = 0.12$; Dark trials: Error at periphery, $P = 0.25$; Search speed, $P = 0.37$; Homing speed, $P = 0.37$); ns: non-significant.

Fig. 1R is now Extended Data Fig. 5 in the manuscript.

Based on these findings, we consider it unlikely that mice used odors as the primary sensory information to return to the home base during dark trials. The correlation between the search path length and homing accuracy during dark trials suggests that self-motion cues were key in guiding the mouse back to the home base. These points are discussed in the section “Path integration contributes to the directional selectivity of lever-anchored firing fields in darkness” of the Discussion.

3. It is good that Extended data Figures 3 and 4 show analyses using the mouse as a unit of analysis. For Figure 3H-K, I think the analysis in the extended data is more correct and should be used in the main figure. For Figure 4, the goal of showing the data distribution at the level of trials is appropriate, but the statistical analysis should still account for dependencies at the level of animals in order to avoid overinflated degrees of freedom (and artificially low p-values). This could be done by adopting a hierarchical approach or by using the mice as unit of analysis (as in the extended data). It is also important for these analyses to report the test statistic and degrees of freedom alongside the p-value.

This issue was raised by both Reviewers (see R2, point 10). We have modified our statistical procedure in many sections of the manuscript to use mice instead of trials (and sometimes neurons) as statistical units (Figures 1, 2, 3, 4, 6, and 7). The trial distributions for behavioral measures (error at periphery, running speed, etc.) are now shown in Extended Data Figs. 3-4, but we report the statistics performed with mice as statistical units.

We also report the number of data points, tests used, and p-values when reporting statistics.

4. Were probes moved between recording days? Are data reported from the same mouse for multiple days? If so, how is double counting of the same neuron avoided?

We thank the reviewer for noticing this omission in the Methods section. We now state that we moved the probes down by approximately 50 μm after each daily recording session. We expect that a small subset of neurons was recorded on consecutive days. It is, unfortunately, very difficult to identify these neurons, given that the probes were moved between sessions, leading to the same neurons having different waveforms across days. In the revised manuscript, we confirmed most conclusions using mice as statistical units. This should ensure that having some neurons recorded over more than one day does not affect the conclusion of our study.

5. Over-inflation of the degrees of freedom for statistical analyses of the neuronal firing is also a potential concern. As presented the analyses implicitly assume that all neurons (or neuron pairs) are independent. This ignores dependencies that arise from neurons being recorded from the same animal and in the same session. I don't think the conclusions are likely to change in this case but it is nevertheless important to assess whether such dependencies exist and if necessary modify the analyses accordingly (see e.g. <https://www.jneurosci.org/content/38/26/5837>).

We have modified several sections of the manuscript to address this point. When analyzing remapping between tasks or between AutoPI trial conditions, we now perform the analysis with mice as statistical units (Fig. 3d and 3g). When linking the activity of the lever-anchored firing fields to search and homing behavior, we also performed the statistical analysis with mice as statistical units (Fig. 6b-g and Fig. 7b). Thus, the main conclusions of the manuscript are now supported by statistics performed on independent observations. In Fig. 5d-g, we opted for using neurons as statistical units because the analysis mainly focused on the information encoded by the neurons, without a strong link to the mice's performance.

6. Could the weak firing rate correlation between foraging and AutoPI conditions be caused by dark trials? What if these are excluded?

This is an interesting possibility that we addressed by assessing remapping between random foraging and the AutoPI task separately for light and dark trials. We observed weak correlations between random foraging and both types of trials. We have modified Fig. 3d to include this analysis.

7. Firing rate similarity result. What was the correlation between first and second half of the foraging session?

The first half of the foraging session refers to the first 15 minutes of foraging, while the second half is from 15 to 30 minutes. The correlation coefficient is reported in the boxplot of Fig. 3d (RF1-RF2, 0.74). We have modified the scheme of the recording session to illustrate the first (RF1) and second (RF2) half of the foraging trial (Fig 3a).

8. I wonder if Figure 4a would be more useful if it showed the same neuron across all four conditions? As I understand it, at the moment it shows separate example neurons for each condition, but this leaves open the question of what each neuron's activity looks like in the other condition.

We opted to show different neurons in each condition because most neurons have a clear firing field only in one or two conditions (now shown more clearly in the new Fig. 3). To illustrate this, we have generated Fig. 2R (see below) for Reviewer 1 showing the four conditions for each neuron.

a**b****c****d**
Fig. 2R. Activity of four pyramidal cells during the four behavioral conditions (Search-Light, Search-Dark, Homing-Light, and Homing-Dark). For each neuron, we show the firing rate map and the trial matrices for the four conditions. Many neurons have a clear firing field in only one of the four conditions.

We decided to show different neurons in the different behavioral conditions to save space and simplify the figure.

9. In testing for whether directional selectivity is influenced by search path length, it would be helpful to show the scores reported in Figure 6b-e as a function of path length (rather than differences). This could be considered first at the level of individual animals. According to the path integration hypothesis these scores should correlate with path length, at least for animals with a reasonable distribution of path lengths. Is this the case?

We have modified Fig. 6b-g to show the scores reported as a function of path length using mice as statistical units. We note that the difference of MVL after short and long search paths does not reach statistical significance when using mice as statistical units ($P = 0.07$). The other differences remained significant.

We also calculated Pearson correlation coefficients between the trial drift of lever-box anchored fields and the search path length for each lever-box-anchored neuron and report the distribution for each mouse (Fig. 2Rb, also Fig. 6f). The median correlation coefficients (one per mouse) were significantly larger than 0 (Fig. 2Rc, also Fig. 6g).

Fig. 3R. a, The distribution of search path lengths for eight mice. b, Distribution of Pearson correlation coefficients between the search path length and the trial drifts for each mouse. c, Distribution of median correlation between search path length and the trial drifts ($n = 8$, Wilcoxon signed-rank test, $P = 0.007$). d, The correlation between search-path/trial-drift correlation coefficients and variance of search path lengths during the recording session for each lever-anchored field.

Related to this, for the analysis in Figure 6, is there are a large range of path lengths for all animals? If not, then I wonder if the analysis is obscuring a larger true effect; if some animals

have little variation in their path lengths then their difference scores (Figure 6b-e) would on average be close to zero.

The distributions of search path lengths were relatively similar between mice (Fig. 3Ra). The distribution of correlation coefficients between search path length and trial drift is shown in Fig. 6f-g to give information about the between-animals variability. We did not find a relationship between the standard deviation of the search path length and the correlation coefficient (search path length - trial drift) (Fig 3Rd).

10. Similar comments for Figure 7b as for Figure 6b-e.

We have modified Figure 7b and now report the effects using mice as statistical units. Mean vector lengths and matrix correlations were significantly larger for accurate than inaccurate trials. The distribution of homing error for each mouse is shown in Fig. 4Ra.

Fig. 4R. a, The distribution of homing error at periphery for eight mice. The dashed line represents the mean of the homing error for each mouse. b, Distribution of Pearson correlation coefficients between the homing error at periphery and the trial drift for each mouse. c, Distribution of correlation for the homing error at periphery and the trial drift ($n = 8$ mice, Wilcoxon signed-rank test, $P = 0.039$). d, Absence of linear relationship between the homing-error/trial-drift correlation coefficients and the standard deviation of homing error for each lever-anchor field.

11. Extended Data Figure 7. It could be helpful to briefly make clear the rationale for applying the cross-correlation rather than simply using the peaks of the shifted trials (b).

Our estimate of the firing rate of a neuron as a function of direction around the lever for single trials is noisy because it is based on only 1-2 seconds of data. Performing a cross-correlation between this estimate and the idealized tuning curve of the neuron smooths the single-trial estimate, potentially giving us a more reliable estimate of the trial drift. This rationale was added to Extended Data Fig. 9.

12. The use of 'several neurons' when summarising the work is a little vague (e.g. abstract and discussion). Perhaps indicate the proportion?

We thank the Reviewer for reporting our vague statements. We now give proportions in the abstract and discussion.

Reviewer #2:

This paper reports a study in which hippocampal place cells were recorded from mice performing a continuous homing task, in which they had to search an area for a release lever and then navigate to the home box to retrieve the food made available by the lever. These trials were recorded both in light, when the animals could see the home box, and in darkness when they had to navigate using their internal sense of movement and space (i.e., using path integration). The study addressed two questions: (i) whether the firing patterns present in animals actively engaged in path-integration-based navigation are the same as during other types of navigation, and (ii) whether the activity patterns of spatially selective neurons during path-integration-based navigation are linked to successful navigation. The main findings are (i) that the place cells remapped their firing fields between phase of the task (search vs. homing) and between light and dark; (ii) that around 25% place cells had firing fields that were located in a given direction from the mobile lever at a fixed distance from it, and (iii) that parameters of the firing such as the field's spatial stability and direction relative to the lever correlated with navigation behavior. It is concluded that "the changes in navigational strategy were responsible for the reorganization of hippocampal cell ensembles when access to visual landmarks was changed" and (in relation to the lever-anchored cells) "This observation rules out the simpler hypothesis that all hippocampal neurons fire in a global and stable reference frame that includes the starting point of their journey."

This is a nice behavioral task addressing an issue that has long been of interest to those studying spatial cognition, which is the neural mechanisms that underpin path integration. The data are well analyzed and nicely presented and the manuscript is well-written. However I have some concerns about whether (a) the conclusions are fully supported by the data, and (b) the extent to which the findings are novel. These are as follows.

1. The first major conclusion is that there is remapping between the search phase and the homing phase of the task. This is supported by the data illustrated in Fig. 3 where correlations are made between cell-pairs, with the decline in correlation across task phases used to support the notion that the cells remapped. However this analysis is confounded by the change in behavior between these task phases, such that it is difficult to compare place fields since their sampling may not have been equivalent. For example, a place cell whose field was traversed during search but not homing would change its correlated firing in the two phases, even if its field remained the same. Thus, I am not convinced that remapping is convincingly shown here (the single illustrative examples are also not so convincing, and also highlight the uneven sampling). A better way to do this would be to match trajectories run for run between the two conditions, and ideally to use a many-cells correlation (population vector) rather than restricting to cell-pairs.

Reviewer 2 is correct in pointing out that the sampling of the arena varies widely depending on the conditions (light Vs. dark and search Vs. homing). We have changed our approach to show more convincingly that there is a remapping between tasks and task conditions.

In the previous version of the manuscript, we used the stability of the firing rate associations to provide evidence for remapping. We now use 2D firing rate map instead of firing rate associations as a basis of the analysis presented in Fig. 3. When considering remapping between trial conditions on the Autopi task, we now limit the analysis to a portion of the arena that is relatively well covered in our four behavioral conditions. These modifications ensure that the remapping is not due to stable firing fields being traversed only during one condition.

We added spike-on-path plots for all example neurons to give the reader a better idea of the raw data from which the firing rate maps are generated. These provide additional evidence for our claim that there is remapping between trial conditions. More information regarding these “example” neurons is found in Extended Data Fig. 11.

We also performed many-cells correlation analysis from the firing rate maps. We limited the analysis to the area well covered by the mice in the four conditions (zone of interest in Fig. 3e). For each condition, we created a 3D stack of firing rate maps, including all recorded pyramidal cells. We obtained correlation coefficients by either correlating the pairs of stacks (flattening the 3D arrays to 1D arrays and calculating a correlation coefficient per comparison) (Fig. 5Ra) or by performing correlations that included only the population vectors for a given position, repeating the process for each spatial bin of the map (Fig. 5Rb). This analysis led to a similar pattern of results as with analysis based on pairs of neurons. This data is now in Extended Data Fig. 7.

Fig. 5R. Remapping in the zone of interest on the arena quantified by population vector analysis. a, We created two 3D stacks of firing rate maps for each comparison including all pyramidal cells. The correlation coefficients were obtained by flattening the 3D stacks to 1D arrays and performing one Pearson correlation per comparison. b, We created two 3D stacks of firing rate maps for each comparison. A Pearson correlation was performed at each spatial bin of the firing rate maps (4x8, 5x5 cm bins). The boxplot shows the distribution of r values (from 32 spatial bins) when comparing different behavioral conditions (SL: Search-Light, HL: Homing-Light, SD: Search-Dark, HD: Homing-Dark). The trials of each condition were divided into two independent sets of trials (e.g., SL1 and SL2) to allow within-condition comparisons. Correlations between different conditions (e.g., SL-HL) were all significantly lower than those within conditions (e.g., SL1-SL2) ($n = 32$ spatial bins per comparison, Mann-Whitney U test, all $P = 6.5 \times 10^{-12}$).

1b. In any case, even if there is remapping, such changes have been shown before in related situations (e.g., Markus et al who saw remapping between foraging and goal-directed search, and Geva-Sagiv et al who saw it between visual and echo-location navigation). Therefore it isn't clear that such a finding would be informative about path integration per se, so much as context processing more generally.

We agree with Reviewer 2 that remapping following a change in task is not new. However, one of our original motivations for performing this study was to establish whether there would be a single hippocampal representation of the arena across the four behavioral conditions. On one hand, it could be argued that we could observe remapping between random foraging and the AutoPI task and between the four conditions of the AutoPI task. For instance, Markus et al. reported remapping between random foraging and more stereotyped goal-directed behavior. On the other hand, it has also been reported that place fields remain stable when lights are turned on and off during open field exploration (McNaughton et al., 1989; Quirk et al., 1990; Markus et al., 1994). Thus, we believe the remapping analysis is an important step in characterizing the firing patterns of hippocampal neurons during a homing task.

2. The second main question is whether the firing patterns of the cells are "linked" to successful navigation. The implication is made that such a linkage would imply a causal connection in which the cells drive the behavior. For example, in the Discussion they say "our results demonstrate that object associated firing fields encode information relevant for navigation. We hypothesize that these lever-box-anchored firing fields contribute to the homing behavior of the animal". However they acknowledge themselves immediately afterwards that this remains speculative – it could be that the behaviour informs the fields, or that another factor influences both behaviour and field location (the head direction signal being an obvious candidate). Thus, the results do not answer the second question either.

Reviewer 2 is correct that our findings do not show causality between cell activity and behavior. Our study is correlative, and we aim to establish the firing patterns associated with or linked to successful navigation. We have rewritten the concluding statement of the Abstract and sections of the Introduction and Discussion. We now use "associated with" or "predict," when describing the firing patterns correlated to homing behavior. In the Discussion section, we speculate about the possible role of the hippocampus in homing. We clearly state that no causality can be inferred from our results: *"Although no causality can be inferred from the current study, we hypothesize that these lever-box-anchored firing fields contribute to the homing behavior of the animal."* We also state that *"Studies in which the activity of hippocampal neurons is manipulated at different moments during a homing task will be needed to test whether the activity patterns observed in our study are causally linked to homing behavior."* These changes should ensure that no causality is inferred from our results.

3. The third main finding is of cells that anchor their firing to the mobile lever rather than to the boundaries of the environment. This is an interesting finding but not novel: The phenomenon of place cells attaching to mobile objects at a characteristic direction has been reported previously by Gothard et al., Rivard et al and Deshmukh and Knierim.

We agree that object cells and object-vector cells have been described before, and the relationship between our findings and the studies of Gothard et al., Rivard et al., and Deshmukh and Knierim are discussed. The main contribution of our study is to show that the lever-box-anchored (or object-vector) fields are associated with homing accuracy. To the best of our knowledge, a link between object-anchored firing fields and behavior has not been shown before. In the Introduction, we now discuss the work of Gothard et al. to provide a stronger link between previous work and our findings.

Overall, then, I feel that some more analysis is needed, the conclusions need to be modified, and there needs to be stronger linkage to previous work and the fact that the present work yields more of a replication than completely novel findings (which personally I find a strength of the paper, given the current focus on reproducible science).

Other comments:

4. Abstract: “These results demonstrate how neurons with object-anchored firing fields contribute to navigation.” Is a little too vague to be a useful conclusion, and also makes the incorrect assumption discussed above about the causal link in these phenomena – these fields do not necessarily contribute to navigation (although they might).

We have modified this statement to make it more precise and avoid any reference to causality: “Our results demonstrate that the activity of hippocampal neurons with object-anchored firing fields can predict homing behavior.”

5. L64+ “Among spatially selective neurons, hippocampal place cells are likely to contribute to H-PI. Indeed, hippocampal lesions affect H-PI, and place cells can remain spatially selective when external landmarks are removed... implying that a path integration process can control place cell activity” There is a similar causal direction issue here. The first statement is about how hippocampal cells could affect PI, and it doesn’t follow therefore that PI affects the cells.

We have modified this section. “Hippocampal place cells are part of a neuronal circuit processing self-motion cues. For instance, place cells can remain spatially selective when external landmarks are removed, implying that a path integration process can control place cell activity.”

6. L418 – this section began with extensive discussion of the distance between the firing of the cell and the lever but left out the other obvious point that this distance was at a fixed direction – the cell thus fired with a vectorial relationship to the lever. It is very confusing to have distance and direction separated out like this and I suggest to combine and just make the point about the vector. These are clearly “object-vector cells” and since this terminology already exists it would be best to use it for clarity.

We have considered this point extensively and even discussed it with Prof. James Knierim, who first coined the term “landmark-vector cells” for place cells that encode spatial locations as a vector relationship to local landmarks.

In the manuscript, we treated distance and direction separately because distance coding was measured from the search and homing paths, and the direction of the fields was measured when the animal was directly next to the lever.

We would prefer not to label the neurons with firing fields anchored to the lever as landmark/object vector cells for a few reasons.

1. We do not have a clear vector for each field. For each field anchored to the lever, we either have the direction or length of the vector but not both. During the search and homing path, we can estimate the length (distance) of the vector but not its direction. This is because the homing and search paths were predominantly found between the lever and the home base. This is not unlike the linear track experiments of Gothard et al. (1996). For firing fields located directly around the lever box, we can estimate the direction of the vector but not its length. We would prefer not to label our neurons as landmark/object vector cells given that we can't really define a 2D vector for each field.
2. There are important differences in the percentage of landmark/object vector cells reported by Deshmukh et al., 2013 and what we observed during the AutoPI task. Knierim estimates that approximately 5% of pyramidal cells are landmark vector cells during open-field exploration, whereas we estimated that 24 % of pyramidal cells had a lever-anchored firing field.
3. In a preliminary experiment, we found that the firing of grid cells is anchored to the lever box when the animal is around the lever. These same grid cells have firing fields anchored to the room coordinate system at the beginning of the search path.

For these reasons, our current view is that several functional cell types (at least place cells and grid cells) can have firing fields anchored to the lever. This does not define a cell type per se but rather reflects the reference frame controlling the activity of a group of neurons at a given time. We would therefore prefer referring to these neurons as “neurons with lever-anchored firing fields”. Labeling them landmark/object vector cells implicitly implies that they would fire in reference to objects during open-field exploration (experiments like Deshmukh et al., 2013), which is likely not the case for most of them.

In the manuscript, we use the term “lever-box anchored firing fields”.

We treated distance coding and direction coding around the lever separately because we do not have a measure of distance and direction selectivity for a given field. We now emphasize that one section is about coding for distance to the lever box during search and homing behavior, whereas the other section deals with fields surrounding the lever box.

7. Place field analysis – we need to see at least some of the spike plots as the heat maps are too uninformative as to what is going on. The examples in Fig. 5A are great and it would be good to see more like this. We also need some illustration of cluster separation quality. One suggestion is to have one clear example of the main phenomenon where we see the cluster, waveform, and spike plot so as to gain a really clear picture of what the underlying data look like.

We have modified Fig. 3 and 4 to show more spike-on-path plots. Moreover, we have added Extended Data Fig. 11 to the manuscript that shows the waveforms, spike-time autocorrelation, a linear isolation plot for the spike cluster, spike-on-path plots, and firing rate maps for all “example” neurons used in the main figures (Figs. 3, 4, 5, 6 and 7).

8. Was there any evidence of the prevalence of lever-anchored fields increasing or decreasing across days? (Or both, in different cells perhaps)

We have estimated the prevalence of lever-anchored fields during the first and last recording days of each mouse.

Fig. 5R. Proportion of pyramidal cells with lever-anchored fields during the first and last recording session of each mouse. There was no significant difference between the first and last recording day ($n = 9$, Wilcoxon signed-rank test, $P = 0.52$).

It is important to note that the recordings were performed in well-trained mice that had performed the AutoPI task with light and dark trials for at least 10-15 days before recordings. We also moved the probes down within the pyramidal cell layer during the recording periods. It would therefore be difficult to know whether the animal history or the exact position of the probes within the pyramidal cell layer influences the prevalence of lever-anchored firing fields. We have not included this information in the manuscript as it is difficult to interpret.

9. The overall conclusion doesn't fully sum up the findings. There were two initial questions and the findings yielded a third observation, so all three of these should be summarized.

We have modified the concluding paragraph of the Discussion section. We now refer to all our main findings.

10. Statistics – The statistics are calculated on a per-cell basis, which is OK for basic physiological properties but not for more cognitively relevant ones as these are, where characteristics of the whole animal (such as its head direction orientation) come into play and introduce dependencies in the data. The statistics therefore need also to be calculated on a per-animal basis. I personally also prefer to see that statistics in the text itself rather than hidden away in a supplementary data table (although I understand the decluttering motivation).

Based on this comment and from the feedback of Review 1, we have added analysis with mice as statistical units to support most of our conclusions (Fig. 1h-k, Fig. 2b-g, Fig. 3d, Fig. 3g, Fig. 4c-d, Fig. 6b-g, Fig. 7b). Trial distributions were moved to the Extended Data.

11. Figures – a general comment: it would be good to have the take-home message of each plot included in the legend. These analyses are very hard to unpick and casual readers will not want to spend the time.

We have modified the figure legends to ensure they start with a clear take-home message.

We are concerned that adding a take-home statement for each figure panel would make the figure legend excessively long and would create considerable overlap with the text of the Results section. Therefore, we opted to add only a few take-home messages for the most important figure panels.

12. Also - it would be good to see the individual data points plotted on the bar charts

We have added individual data points to most plots using mice as statistical units.

13. Fig. 2a – typo – missing g on “homing”.

We have corrected this typo.

14. Fig. 4a – I didn't understand the point being made by this figure. Are the same three neurons being shown in every panel? (I assume so). The figure aims to show that lever-box location modulates place cell activity, but according to my understanding, only the third cell shows this. I think it would help to make this more explicit: show just two cells, one modulated by distance to the start box and one by distance to the lever box, and label these clearly as such.

We have modified Fig. 4a (now Fig. 4b). In 3 of the 4 behavioral conditions (Search-Light, Homing-Light, Homing-Dark), we show 2 neurons: one neuron encoding position and one encoding lever distance. We did not observe clear “lever distance” neurons when the mice searched for the lever in darkness. These firing fields are labeled as “Place field” or “Lever distance field” to make this clear.

As mentioned above I think it would also be helpful to have the spike plots so as to get a clearer picture of what is actually happening. I think naïve readers also need some help in understanding the heat plots; the point being that a start-box-anchored cell will have fixed location in the left-hand plot and variable in the right, whereas a lever-anchored cell will show the reverse. Stepping back from this a bit, I don't find the linearized heat plots very intuitive and it might be easier to understand if the data were plotted as field centroids on a 2D plot, one with the origin centered on the start box and the other with it on the lever box.

We have added a new figure panel to Fig. 4 (Fig. 4a) to introduce the concept of firing fields being affected by the distance to the lever box. This panel shows the activity of two neurons during the search behavior in the light. One neuron fires at a specific position, whereas the other fire at a set distance relative to the lever. We hope that this panel will help the reader understand the data presented in Fig. 4.

In Fig 4b, two different neurons are shown in each of the four behavioral conditions.

15. Fig. 5a I found the colors of the spike plots hard to distinguish (albeit tasteful!).

We have modified the colors of the spike plots to make them easier to distinguish.

The take-home message of this figure (Fig. 5) seems to be the same as the previous –these could be distinguished more clearly (or else combined).

We now have distinct take-home messages for Fig. 4 and Fig. 5.

In Fig. 5b it was not immediately obvious what the lever reference frame means because it hasn't been highlighted that the lever itself is rotationally asymmetric. (in fact, it remains unclear how asymmetric it actually is – would a mouse even notice this?)

The lever was 116 mm long, 82 mm wide, and 80 mm high. We do not know whether the mouse can discriminate this shape from one with equal length and width. Our reasoning was that the direction of the small plastic part pressed by the mouse (13 x 10 mm) could contribute to setting the reference frame.

We have modified the figure legend and the text: “In the Lever reference frame, the reference vector was pointing toward the lever (13 x 10 mm part of the lever box pressed by the mouse).”

REVIEWERS' COMMENTS

Reviewer #1 (Remarks to the Author):

The authors have done a good job addressing the previous comments. This is an interesting study that gives important new insight into the roles of the hippocampus in goal-directed behaviours.

Reviewer #2 (Remarks to the Author):

The authors have done a good job of responding in a detailed way to the comments of both myself and the other reviewer. I am happy to recommend publication